# Deep-learning two-photon fiberscopy for video-rate brain imaging in freely-behaving mice

Honghua Guan[1,7], Dawei Li[2,7], Hyeon-cheol Park[2], Ang Li [2], Yuanlei Yue [3], Yung-Tian A. Gau[4], Ming-Jun Li [5], Dwight E. Bergles [4,6], Hui Lu[3] & Xingde Li [1,2,6✉]

Scanning two-photon (2P) fiberscopes (also termed endomicroscopes) have the potential to transform our understanding of how discrete neural activity patterns result in distinct behaviors, as they are capable of high resolution, sub cellular imaging yet small and light enough to allow free movement of mice. However, their acquisition speed is currently sub-optimal, due to opto-mechanical size and weight constraints. Here we demonstrate significant advances in 2P fiberscopy that allow high resolution imaging at high speeds (26 fps) in freely-behaving mice. A high-speed scanner and a down-sampling scheme are developed to boost imaging speed, and a deep learning (DL) algorithm is introduced to recover image quality. For the DL algorithm, a two-stage learning transfer strategy is established to generate proper training datasets for enhancing the quality of in vivo images. Implementation enables video-rate imaging at ~26 fps, representing 10-fold improvement in imaging speed over the previous 2P fiberscopy technology while maintaining a high signal-to-noise ratio and imaging resolution. This DL-assisted 2P fiberscope is capable of imaging the arousal-induced activity changes in populations of layer2/3 pyramidal neurons in the primary motor cortex of freely-behaving mice, providing opportunities to define the neural basis of behavior.

[1] Department of Electrical and Computer Engineering, Johns Hopkins University, Baltimore, MD 21218, USA. [2] Department of Biomedical Engineering, Johns Hopkins University School of Medicine, Baltimore, MD 21205, USA. [3] Department of Pharmacology and Physiology, School of Medicine and Health Sciences, George Washington University, Washington, DC 20052, USA. [4] Solomon H. Snyder Department of Neuroscience, Johns Hopkins University School of Medicine, Baltimore, MD 21205, USA. [5] Science and Technology Division, Corning Incorporated, Corning, NY 14831, USA. [6] Johns Hopkins Kavli Neuroscience Discovery Institute, Baltimore, MD 21218, USA. [7] These authors contributed equally: Honghua Guan, Dawei Li. ✉email: xingde@jhu.edu

Establishing correlations between the activity of a population of neurons with discreet animal behaviors is a critical step in understanding how the brain encodes motor output. A device capable of real-time activity imaging of a group of neurons with subcellular resolution holds promise for elucidating such correlations. Multiphoton microscopy, along with genetically encoded fluorescent calcium indicators (such as GCaMP), has become essential methods for studying neural circuit dynamics in the brain[1,2]. In vivo two-photon imaging generally requires head-fixation under the microscope objective[3]. Such constraint and associated physical stress inevitably influence neuronal activity and preclude many behavioral assays that require freely moving animals, such as elevated plus maze and social interaction tests[4]. Flexible, compact 2P fiberscopy techniques have been developed by several groups, including ours[5–10], which can potentially allow continuous imaging of brain activity in a freely behaving configuration over a long period of time[11,12]. In our compact fiber-optic scanning 2P fiberscope, the key design elements include a single customized double-clad fiber, a miniature piezoelectric actuator, and a super-achromatic micro objective lens[13]. For functional neuroimaging, the probe is attached to the animal's head. It focuses the femtosecond excitation light to the target brain region, scans the focused beam across the field of view (FOV), and collects the 2P fluorescence from the target to form an image.

However, the ultra-compact design of the imaging probe, limits the choices of beam scanner and imaging optics, and consequently limits the imaging frame rate (usually <5 fps)[7]. The suboptimal frame rate makes 2P fiberscopy subject to motion artifacts induced by normal high-frequency physiological activity (such as heartbeat at 8-13 Hz)[14,15]. An increased imaging frame rate is highly desirable to mitigate motion artifacts[16], which becomes more critical when imaging freely moving animals. In addition, for spiking activities analysis, a high imaging frame rate can significantly improve the accuracy of the resulted firing rate[17], particularly when using fast calcium indicators such as GCaMP6f. Furthermore, a higher imaging frame rate (>15 Hz) is usually required when exploring the dendritic dynamics in living brain[18]. Accordingly, it is essential to improve the frame rate of the 2P fiberscopy technology for functional neuroimaging.

The frame rate of a conventional flying-spot, raster-scanning imaging system is determined by the scanning speed (line scanning rate) divided by the scanning density (line density). The same applies to spiral scanning fiberscopes, where the scanning speed is defined by the number of spirals per second (*i.e.*, the resonant frequency of the fiber-optic scanner[5–7,13]) and the scanning density is defined by the number of spirals per frame (*i.e.*, the sampling density along the radial direction) (see Methods -> Scanning 2P fiberscope system for details). The frame rate can be improved by increasing the scanning speed or/and decreasing the radial scanning density. However, an increased frame rate inevitably affects image quality, as a higher scanning speed sacrifices the signal-to-noise ratio (SNR) due to a shorter pixel dwelling time, and a lower scanning density comprises the imaging resolution.

Here we present a deep neural network (DNN) based solution that significantly improves the imaging frame rate with minimal loss in image quality. This innovation enables 10-fold imaging frame-rate enhancement of fiberscopy, making it feasible to perform vide-rate (26 fps) 2P imaging in freely moving mice with excellent imaging resolution and SNR that were previously not possible. To determine the proper training dataset for high-speed in vivo imaging data, we utilized a two-stage learning transfer strategy to manually generate the intermediate training input and ground truth. Using this DNN-assisted method, high SNR and high imaging resolution neuronal images could be recovered from low SNR, low imaging resolution images acquired at much higher frame rates.

## Results

This work aims to accelerate 2P fiberscopy imaging without degrading imaging quality. In pursuit of this goal, we introduced a DNN-based strategy (see Methods -> Deep neural network (DNN) for details) to recover the associated SNR decrease and imaging resolution loss induced by a high frame rate. We adopted the pix2pix framework derived from a conditional generative adversarial network (cGAN)[19,20]. The method involves a training stage and an inference stage, as shown in Fig. 1. The training stage optimizes the desired DNN model from a given training dataset, which includes the input images and the ground truth (Fig. 1a). During training, the optimizer discriminates the differences between the generated output and the ground truth defined by an objective function and then updates the DNN to minimize the differences. After a sufficient number of iterations, the trained network can be used to enhance the quality of new images (Fig. 1b).

The performance of a DNN is highly dependent on the training datasets, especially the quality of the ground truth. The ideal ground truth images for 2P fiberscopy are those with high SNR and high pixel density (or imaging resolution). For ex vivo imaging, this can be achieved by using frame averaging and high-density sampling, which are commonly adopted when applying deep learning to optical microscopy for biological sample imaging (ex vivo)[21,22]. For in vivo imaging, particularly in freely moving mice, however, we cannot obtain the ideal ground truth, where frame averaging is generally not feasible due to motion artifacts and the dynamic nature of neuron activities.

To address the problem, we introduced two-stage neural networks with the network at the first stage (DNN-1) helping produce approximate ground truth which could be used for training the second stage DNN (DNN-2). The flow chart of our method is shown in Fig. 1c: In the first step, we trained a denoising network (DNN-1) using an ex vivo imaging dataset. Here, an intentionally decreased scanning speed and frame averaging were adopted to optimize the SNR of ground truth. We then applied the trained DNN-1 to an in vivo imaging dataset acquired from three head-fixed mice over different FOVs to generate the approximate ground truth for the second neural network (DNN-2). In the second step, the same in vivo dataset was digitally down-sampled (mimicking images acquired at a low scanning or sampling density) to serve as the training input. By paring these input images with the ground truth generated by DNN-1, we were able to train DNN-2 for both the denoising and inpainting functions (where inpainting aimed to recover the imaging resolution for the images of low sampling density). This trained DNN-2 was then used for enhancing the high-speed, in vivo imaging dataset collected from two different freely behaving mice over different FOVs. It is noted that a given DNN-2 corresponds to one scanning density. We need to train the DNN-2 separately for images collected with a different scanning density.

To train DNN-1, we used the 2P fiberscope to collect a training image set from two ex vivo GFP-immunostained brain slices (from two mice) over 100 FOVs (denoted by training set I, acquired at a frame rate of ~2.0 fps). The images were acquired at a scanning speed of 1650 spirals/sec and a scanning density of 512 spirals/frame (~3X Nyquist density). For each FOV, multiple frames were collected. We then randomly selected one frame as input for DNN-1 and pared it with the 10-frame-averaged image as the corresponding ground truth (equivalent to a slow-scanning speed of 165 spirals/sec, 0.2 fps). Finally, noise was added to the input images to mimic different imaging conditions. The trained

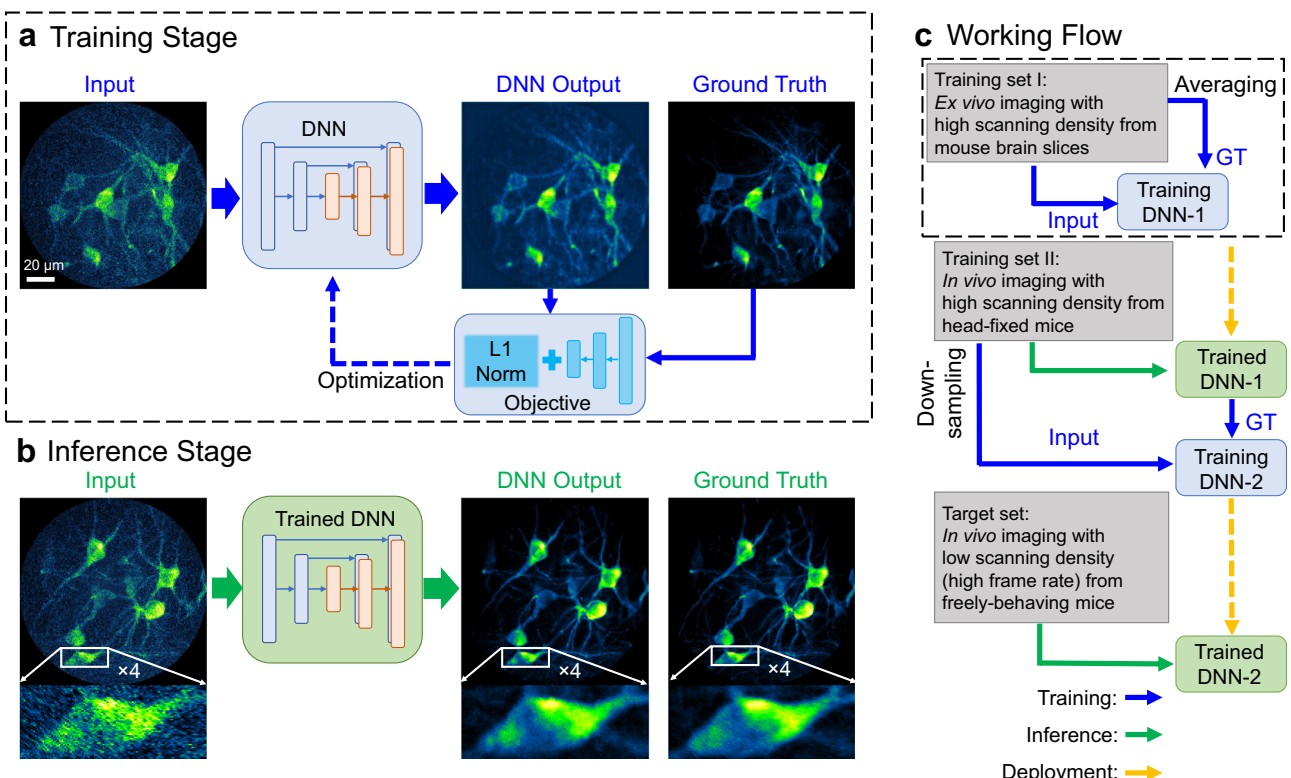

**Fig. 1 Overview of the deep neural network (DNN) based method for video-rate 2P fiberscopy. a** Training stage. **b** Inference stage. **c** Workflow of the two-stage DNNs' training protocol. GT ground truth.

DNN-1 then served as a customized denoiser to improve the SNR of two-photon images collected in vivo with the animal head fixed for generating the ground truth for DNN-2 as described in detail below.

To train DNN-2, we collected images (denoted by training set II) from head-fixed awake mice that expressed calcium indicator GCaMP6m in motor cortex neurons (see Methods -> In vivo imaging for details) at a frame rate of ~3.3 fps with a higher scanning speed (3,360 spirals/sec) but at the same scanning or sampling density (512 spirals/frame) as training set I. It is noted that the head-fixed configuration ensured less motion artifacts than freely moving during data acquisition, and it was more suitable for acquiring training dataset. We enhanced the SNR of training set II by using the trained DNN-1, and the output (i.e., the denoised training set II) served as the intermediate ground truth. Next, for each frame in training set II, we digitally down-sampled the image by decreasing the number of spirals per frame (i.e., down-sampling along the radial direction). The down-sampled images served as the intermediate input, which along with the corresponding intermediate ground truth formed the training dataset for DNN-2. The as-trained DNN-2 aimed to rescue both the SNR and imaging resolution for images collected in vivo at a higher frame rate (i.e., with a higher scanning speed and a lower scanning density). To confirm the feasibility of the trained DNN-2, we manually selected and labeled a set of in vivo images collected from freely behaving mice at a high scanning density (512 spirals/frame) as the testing dataset (see Supplementary Fig. 1 for more details) and quantitatively compared the quality of the DNN-2 output images with their corresponding in vivo ground truth. The results confirm that the as-trained DNN-2 works properly when applied to in vivo freely behaving images with a low scanning density. The trained DNN-2 enabled the detection of fine features that were consistent with the ground truth but difficult to resolve in the original images.

**Performance characterization of DNN-1 for SNR enhancement of in vivo 2P imaging**. We first evaluated the performance of the trained network DNN-1 for improving image SNR by applying the network to testing set I (collected from head-fixed GCaMP6m-expressing mice with an imaging speed of 3,360 spirals/sec and an imaging density of 512 spirals/frame). Figure 2a shows a representative image and its corresponding DNN-1 output (Fig. 2c). The output image exhibits clearly discernable somas and dendrites with its background noise significantly suppressed. For example, the SNR of the selected neuron (Fig. 2b, d) increases from 5.18 to 9.78 dB (see Methods -> Data processing for details) after DNN-1 enhancement (see Supplementary Video 1 for details). Besides image quality improvement, it is critical to assess the potential impact of DNN-1 on the dynamic neuronal GCaMP signals. We thus identified 17 representative active neurons within a given FOV (Fig. 2e) using a well-established post-processing pipeline[23] and compared the normalized time-dependent GCaMP fluorescence intensities ($\Delta F/F$) of the neurons before and after DNN-1 enhancement (Fig. 2f). Results show that the calcium signal derived from the DNN-1 output was highly consistent with the input. We also quantified the differences of the GCaMP fluorescence signals for each neuron between the raw and DNN-1 output images. This DNN-induced signal error was calculated by normalized root-mean-square error (NRMSE, see Methods -> Data processing for details), and we found that the average discrepancy was around 3% (Fig. 2g). This confirms that the introduction of DNN does not modify the temporal neural dynamics, which is critical for accurate functional brain imaging.

It is worth mentioning that the primary purpose of DNN-1 was to restore "neuronal" features from noisy backgrounds. Besides denoising, DNN-1 also performed the end-to-end inverse mapping to preserve sharp edges and fine structural details rather than over-smooth textures. To achieve optimal image restoration, the training and testing dataset should share the same

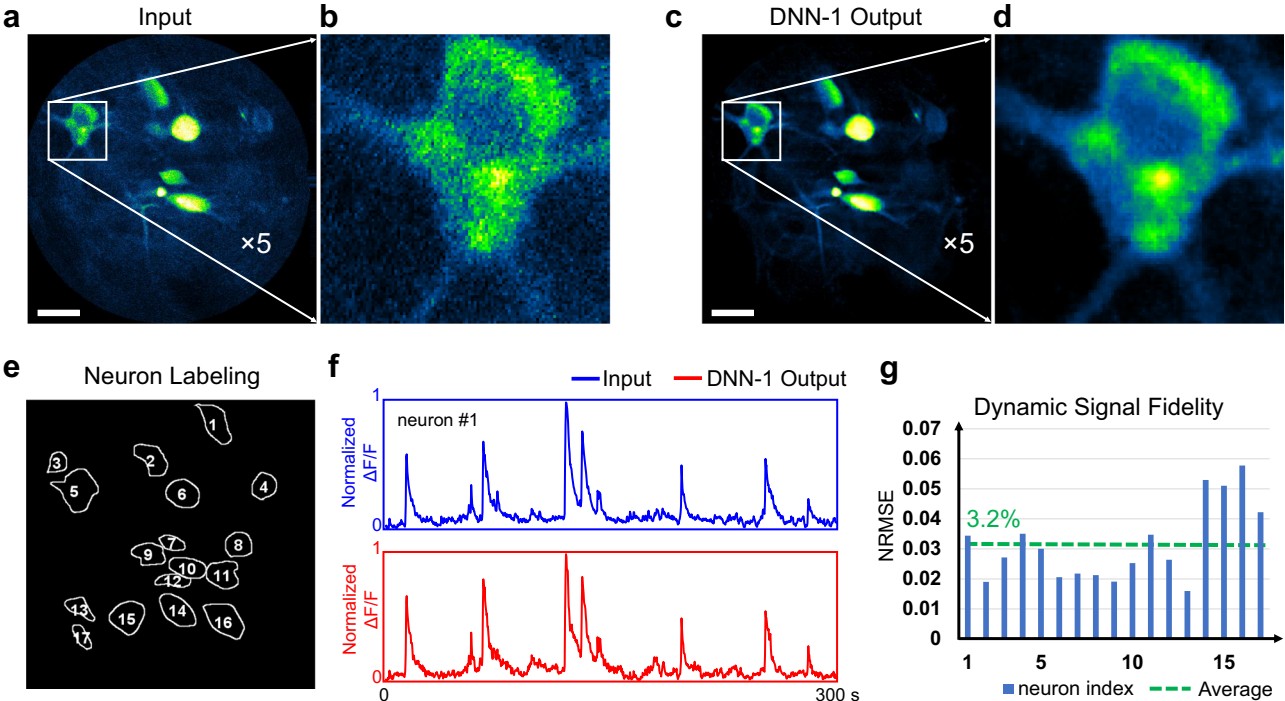

**Fig. 2 Performance of DNN-1 for SNR enhancement of in vivo head-fixed 2P fiberscopy images. a** Representative input image along with the magnified view (**b**) of the selected region. Images were acquired at a scanning speed of 3,360 spirals/sec and an imaging density of 512 spirals/frame. **c** Corresponding DNN-1 output image of (**a**) along with the magnified view (**d**). **e** Neuron identification by using a post-processing pipeline. **f** Normalized time-dependent fluorescence intensity ($\Delta F/F$) of a selected neuron before and after DNN-1 enhancement. **g** Dynamic signal fidelity measured by normalized root-mean-square error (NRMSE) for each neuron. The differences of calcium signals between the DNN-1 output and the raw data are small (with an average discrepancy around 3%). Scale bars in (**a**, **c**): 20 μm.

specific image features or cell types, which include soma, associated axons, and dendrites in this work. Applying a trained DNN-1 to images of an unknown cell type would generate unwanted artifacts, and one example is presented in Supplementary Figure 2.

**Performance characterization of DNN-2 for resolution recovery with down-sampled data.** As it is not feasible to define the proper ground truth of a sufficient number (e.g., 100 or more) of independent images for high-speed functional imaging in freely behaving animals, we introduced a two-stage learning transfer strategy to generate the intermediate training dataset and developed a final network (DNN-2). The DNN-2 enabled imaging resolution recovery for the down-sampled images, compensating for the side effects induced by lower scanning densities. In addition, the DNN-2 inherited the function of SNR improvement from DNN-1; thus DNN-2 was trained to achieve (1) image denoising and (2) pixel inpainting (up-sampling) simultaneously. To quantify the reconstruction accuracy for spatial structural details at different scanning densities, we separately trained the DNN-2 network with in vivo images of different (low) scanning densities. These images were obtained by digitally down-sampling the images in the previous training set II (acquired from head-fixed mice in vivo with a scanning speed 3360 spirals/s and a scanning density of 512 spirals/frame), with a different down-sampling factor M along the radial direction. For example, $M = 2$ means a scanning density of 256 spirals/frame, $M = 4$ means 128 spirals/frame, and so on. The corresponding testing set II (Fig. 3a, acquired under the same conditions as the previous training set II but from different mice) was first enhanced for SNR by the trained DNN-1, and the denoised images (Fig. 3b) were defined as the reference for assessing reconstruction

accuracy. We then spatially down-sampled the images in the testing set II along the radial direction by a factor of M (Fig. 3c, top row) and applied the corresponding trained DNN-2 (with the same M-factor) to these down-sampled images. The DNN-2 output images exhibit continuous and clearly resolved neuronal structures (Fig. 3c, bottom row) in close agreement with the reference image (see Supplementary Video 2 for details). The results reveal that fine spatial details can be recovered from a reduced number of pixels of the original images with the help of DNNs, suggesting that images can be acquired at a much lower scanning density, providing a significant boost for the imaging frame rate (e.g., by a factor of M). Notably, the capability of DNN-2 for recovering the fine spatial details deteriorates as the down-sampling factor M increases (Fig. 3d), implying a trade-off between the reconstructed image quality and scanning density (or imaging frame rate). To quantify this trade-off, we calculated the multi-scale structure similarity (MS-SSIM)[24] index and the normalized root mean square error (NRMSE) of the reconstruction against the reference image (Fig. 3e). We found that for a down-sampling factor $M \leq 8$ (corresponding to a data acquisition frame rate of ~26 fps), the reconstruction for spatial details remains accurate with an MS-SSIM better than 85% and an NRMSE <3%.

In addition, we introduced another protocol to train the DNN-2. We initialed the generator and discriminator of DNN-2 with the pre-trained weights (inherited from DNN-1, denoted as "Pretrain") rather than random weights (denoted as "Scratch" which was used in the above DNN-2 training). Compared with "Scratch" scheme, the "Pretrain" configuration had faster convergence at the beginning. The two training methods were similar to each other after a certain number of epochs (~40) and the loss curves became nearly identical (see Supplementary Fig. 3 for more details). If the training dataset gets significantly larger

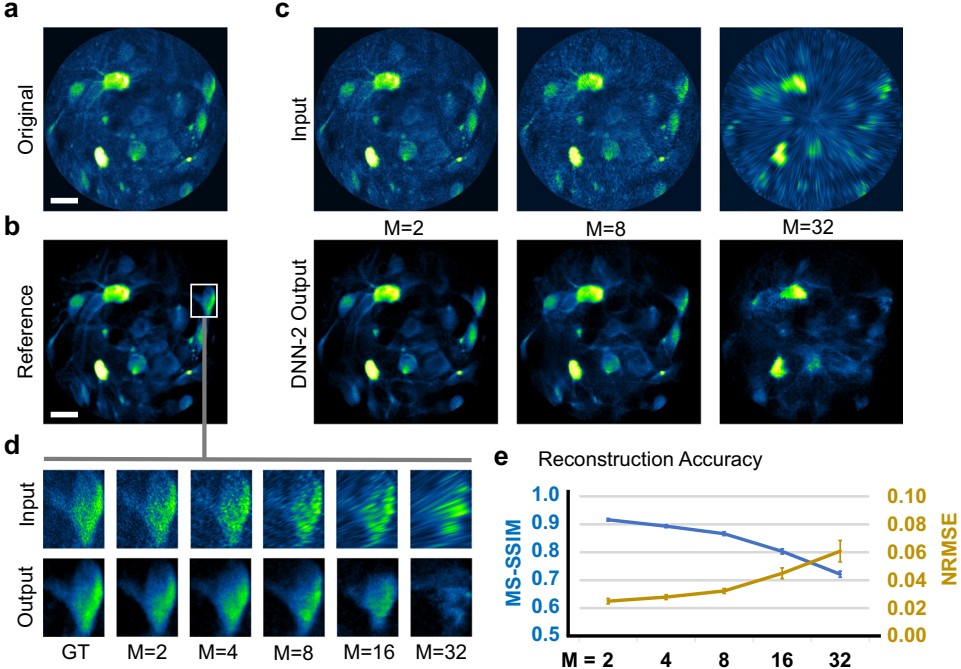

**Fig. 3 Performance of DNN-2 for resolution recovery for in vivo head-fixed 2 P images acquired at various scanning densities (from 512 spirals/frame to 512/M spirals/frame, _M_ = 2–32).** Each scanning density requires a separately trained network. **a** Representative original image (with a scanning density of 512 spirals/frame) and **b** the corresponding DNN-1 output image which served as the reference for quantitative image quality comparison. **c** Top-row: in vivo images of a lower scanning density as the input for testing DNN-2. These images were obtained by digitally down-sampling the original image by a factor of _M_ along the radial direction; bottom-row: DNN-2 output images. **d** Structural details of a selected region of interest with respect to different _M_ factors. **e** Reconstruction accuracy in terms of the multi-scale structure similarity (MS-SSIM) index and normalized root-mean-square error (NRMSE) for the entire testing image set with respect to the down-sampling factor M. The data are presented as mean values (data points) ± standard deviations (error bars). The measurements were made over the whole image stack with a total number of images _n_ = 800. Scale bars in (**a**, **b**): 20 μm.

and the computation cost becomes the major constraint, the "Pretrain" scheme will be an effective alternative.

**Video-rate recording of brain activity in freely behaving mice.** To assess the performance of this DNN-based strategy, we performed functional imaging experiments in freely behaving mice. The mice used to perform freely moving imaging were different from those head-fixed ones used for collecting the in vivo data for training DNN-2. The schematic of the imaging setup is shown in Fig. 4a (see Methods -> In vivo imaging for details). To mitigate interference of normal physiological activity, such as respiration and heartbeat (~10 Hz in conscious mice)[25], we set the down-sampling factor M to 8 during real-time recording (corresponding to a scanning density of 64 spirals/frame with a given scanning speed of 3360 spirals/sec), and acquired 2P images at ~26 fps, providing a good compromise between image quality and frame rate (Fig. 3e). Comparing to the raw images, the output images from the trained DNN-2 exhibited improved SNR and imaging resolution (Fig. 4b, c). Figure 4d shows three representative time-series images where different neurons were active at different times (see Supplementary Video 3 for details). With our DNN-based method, neuron somas and dendritic structures could be clearly recognized during video rate image acquisition After DNN-2 enhancement, we identified and segmented 21 active neurons (Fig. 4e) using the post-processing pipeline, the calcium dynamics for which are plotted in Fig. 4f. In addition, calcium changes within each neuron could also be tracked at high temporal resolution. As shown in Fig. 4g, h, the transition of neurons from quiescence (baseline) to active states could be resolved with a temporal resolution of ~38 ms, suggesting the DNN-assisted 2P fiberscopy system is suitable for in vivo imaging of faster calcium

indicators such as GCaMP6f$_u$[26]. Thus, this approach enhances the temporal resolving capability of 2P fiberscopy, allowing neuronal activity to be monitored under physiological conditions in freely moving animals, providing a more accurate measure of cellular dynamics during discrete behaviors.

**Discussion**
Decoding the neural activity patterns that underlie behavior remains a central goal in neuroscience. Here, we describe advances in 2P fiberscopy that enable high-speed (~26 fps) imaging in freely behaving animals, providing the means to assess neural activity in defined circuits as animals engage in normative behaviors. Video-rate imaging was achieved by increasing the scanning speed and decreasing the scanning density during data acquisition in conjunction with the assistance of DNNs. Compared with existing 2P fiberscopyconfigurations[6,7,13], we increased the frame rate by over 10-fold without compromising signal-to-noise ratio and imaging resolution. This significant improvement in frame rate overcomes a critical bottleneck of 2P fiberscopy and enables it as a promising tool for functional neural imaging studies.

Our DNNs achieved simultaneous image denoising and pixel inpainting. For image denoising, many classical digital image processing methods are widely used and have made great contributions, such as non-local mean filter[27], anisotropic filtering[28], and wavelet-based methods[29]. Usually, these methods require prior knowledge about the noise model of the images and a rational estimate about the noise level. In comparison, deep learning-based methods are advantageous. DNN can effectively figure out the system noise distribution and serve as a highly customized denoiser without the need for complex analyses of the noise model. Therefore, DNN

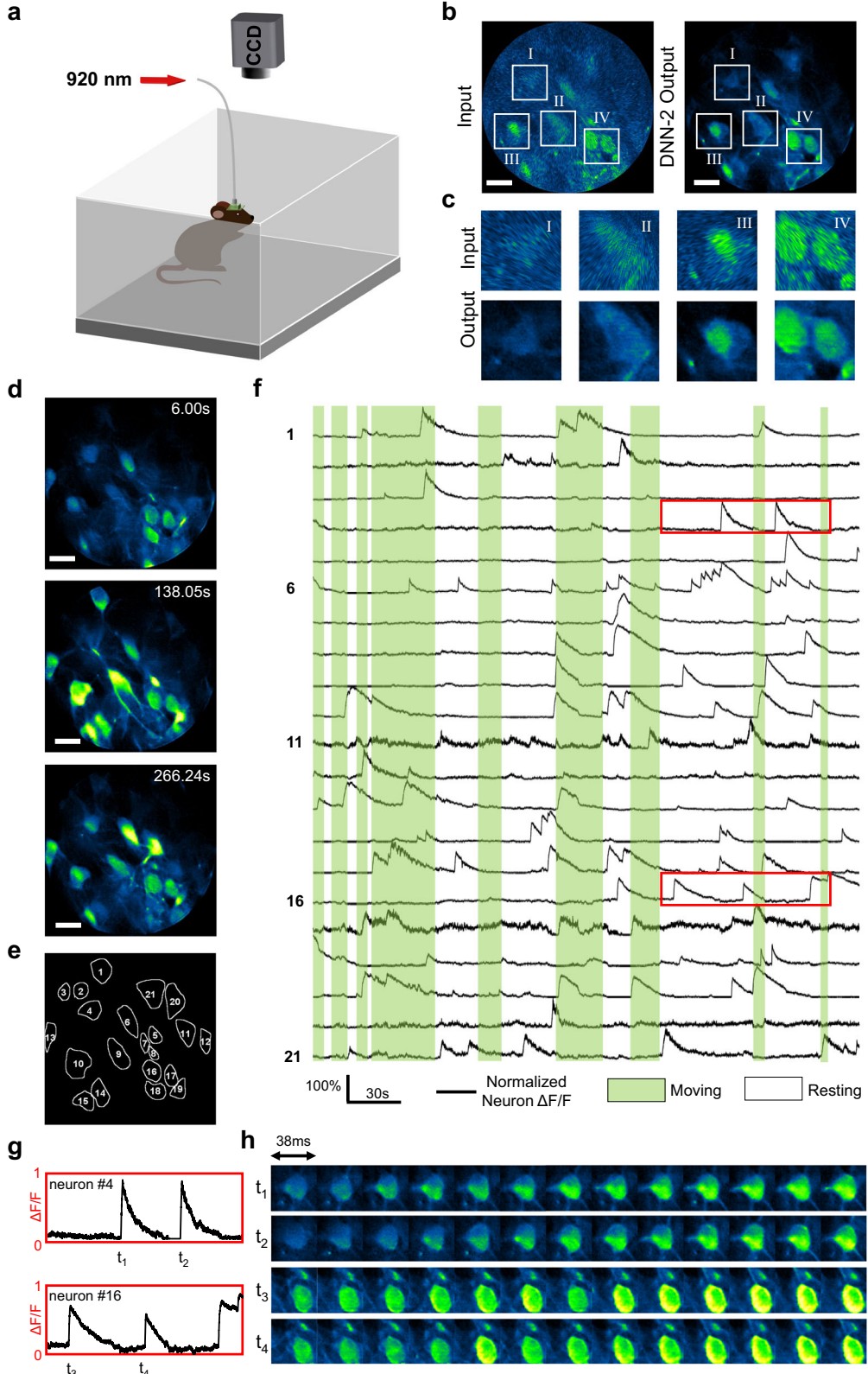

**Fig. 4 Application of the trained DNN-2 to high-speed (26 fps) 2P fiberscopy brain imaging in freely behaving mice. a** Imaging setup schematic of freely behaving mice. **b** Representative high-speed in vivo 2P images before (left, as input) and after (right, as output) DNN-2 enhancement. **c** Comparison between the DNN-2 input and output images for fine structures marked in the regions of interest (ROIs) I–IV in (**b**). **d** Representative dynamic images of neuron activities recorded at different time points. **e** Segmentation masks for the 21 neurons identified within the FOV obtained with the post-processing pipeline23. **f** Normalized calcium (GCaMP6m fluorescence) signals along with behavior registration for the 21 neurons. **g** Zoomed-in temporal dynamics for two randomly selected time windows as marked in (**f**). **h** Spiking process of neurons corresponding to different time points shown in (**g**). Scale bars in (**b**, **d**): 20 μm.

shows better performance, especially when processing some fine structures. One example of qualitative and quantitative comparisons between some traditional imaging denoising methods and our reported DNN-1 is shown in Supplementary Fig. 4 and Supplementary Table 1. Image inpainting is another area where DNN prevails[30,31]. Given similar morphological characteristics of neurons, Nyquist sampling of the entire FOV is unnecessary. With only a fraction of the image sampled, DNN can figure out the rest of the pixels based upon its prior knowledge (training). While there is a common concern about the fidelity of DL-based image processing methods, we have demonstrated that our proposed method is very unlikely to eliminate features/events or create features/events that do not physically exist in the raw data, and one representative comparative study on features in the raw image and the corresponding DNN-2 processed image is shown in Supplementary Fig. 5.

The DL-based method not only improves the visual quality of the images, but also facilitates calcium dynamics analysis. One example is shown in Supplementary Fig. 6. The results demonstrate that the DNN-2 processed images yield better accuracy and sensitivity for segmenting neurons with a weak SNR. Another example is shown in Supplementary Fig. 7. Owing to improved image quality (SNR and spatial imaging resolution), the DNN-2 processed images lead to much improved segmentation of fine structures such as the dendrites. In addition, motion correction would enjoy an improved image SNR[32], which can be particularly critical when imaging freely behaving animals.

The performance of DNN depends on the quality of the training dataset, especially the ground truth. The critical challenge is to generate a proper training dataset suited for high-speed functional imaging in freely behaving mice. Since the ideal ground truth under this imaging condition cannot be obtained directly, one simplistic choice is the frame-averaged ex vivo images as the ground truth. However, there is a big gap between low-speed high-scanning density ex vivo imaging and high-speed low scanning density in vivo imaging. If we only train the DNN with the ex vivo images to perform simultaneous image denoising and pixel inpainting directly (i.e., following a single-stage Pix2Pix training procedure[19]), the DNN outputs become suboptimal when applying the trained DNN to in vivo images collected from freely moving mice. One example is shown in Supplementary Fig. 8. The results demonstrate that our proposed two-stage training procedure is a more feasible solution to handle high-speed in vivo images. In this application, the transferability of the DNN is based on the structural similarity for given cell types (such as neurons, axons, and dendrites) under ex vivo and in vivo conditions. This two-stage DNN-based method can be potentially applicable and valuable to other imaging modalities for increasing the frame rate while avoiding image quality loss.

The current imaging frame rate (~26 fps) corresponds to about 30 ms per frame. This frame rate is clearly not fast enough for direct assessment of action potential which would require a time resolution at the millisecond level (corresponding to a frame rate of several hundred frames/second). The current imaging speed of 2P fiberscopy is physically limited by the scanner. A faster scanner, combined with our deep learning-based approach, may increase the frame rate beyond that achieved here. A further improvement in temporal resolution would enable fast optical recording of sensory-evoked dendritic calcium signals[33], theta oscillations in individual neurons[34] and fast-spike in parvalbumin (PV)-positive interneurons in vivo[35].

## Methods

**Ethical statement**. All the animal housing and experimentation procedures were performed under the standards of humane animal care described in the National Institutes of Health Guide for the Care and Use of Laboratory Animals with protocols approved by the Institutional Animal Care and Use Committees at the George Washington University and the Johns Hopkins University.

**Scanning 2P fiberscopy system**. The details of our 2P fiberscopy system have been reported previously[5–7]. In essence, a femtosecond excitation beam at 920 nm from a Ti:sapphire laser was pre-chirped by a pair of gratings and then coupled into the single-mode core of the double-clad fiber (DCF) used in the fiberscope with an output power set at ~30 mW. The fluorescence signal collected by the micro-objective lens at the distal end of the fiberscope was passed through the DCF core and inner clad to the proximal end and then directed to a photomultiplier tube (PMT). 2D excitation beam scanning was performed by a piezoelectrically actuated fiber-optic scanner. Amplitude-modulated sine and cosine waveforms with a frequency close to the mechanical resonant frequency of the fiber-optic cantilever were applied to the two orthogonal pairs of the electrodes to produce an open-close spiral scanning pattern. The entire assembled probe weighed 0.6 g with diameter of 2.8 mm. The imaging system schematic is shown in Supplementary Fig. 9.

The frame rate of fiberscopy imaging equals to the duty ratio (usually ~50–80%) times the scanning speed (the number of spirals scanned by the fiber cantilever per second) and then divided by the scanning density (the number of spirals per frame). The scanning speed determines the data acquisition time for one spiral, which in turn governs the image signal-to-noise ratio. The scanning density determines the pixel number along the radial direction per frame, and a higher scanning density means a smaller separation between two adjacent spiral scans, which impacts the imaging resolution. It is noted that the pixel number along the circumferential direction (i.e., the pixel number per spiral) is only controlled by the sampling rate of the data acquisition hardware, which does not affect the imaging speed or frame rate.

### Ex vivo imaging

*Animals*. We used C57BL/6J-Tg(Thy1-GCaMP6f)GP5.5Dkim/J (RRID: IMSR_JAX:024276) mice of 20- to 24-week old for this experiment. Mice were maintained on a 12 h light/dark cycle, and food and water were provided ad libitum.

*Immunofluorescence*. We intraperitoneally administered mice with a euthanasia dose of pentobarbital (100 mg/kg body weight). Once in deep anesthesia, animals were perfused with phosphate buffer saline (PBS) followed by cold 4% paraformaldehyde (PFA). Brains were harvested, post-fixed in 4% PFA at 4 °C overnight, cryoprotected in 30% sucrose, and sectioned into 50 μm thick slices on a freezing microtome (Leica SM 2010R). Free-floating sections were washed in PBS, incubated for 1 h in a blocking solution (0.3% Triton X-100 & 5% NDS), and kept at 4 °C overnight with primary antibodies (chicken anti-GFP, Aves Labs, 1:4000) in 0.3% Triton X-100 and 5% NDS. On the day after, sections were washed with PBS, incubated for 2 h at room temperature with secondary antibodies (donkey anti-chicken, Alexa Fluor 488, 1:2000) in 5% NDS, washed again in PBS, mounted on slides and coverslip sealed with mounting medium (Aqua-Poly/Mount, Polysciences #18606-20)[36].

*Ex vivo imaging*. The immunostained sample slide was immersed in the deionized water through entire imaging. The 2P probe was mounted on an XYZ linear stage and then gently positioned against the cover glass of the sample slide. Once the focal plane was determined, the probe was moved laterally (in the X-Y plane) to search for different FOVs and collected imaging data.

### In vivo imaging

*Animals*. We used male mice carrying *Camk2-cre* allele on the C57/B6 for this experiment. The strain was obtained from the Jackson Laboratory (JAX#005359). Animals were given ad libitum access to standard mouse chow and water, housed 4 to 5 per cage in a controlled room of temperature (23 ± 1 °C) and humidity (50 ± 10%) with a 12 hr light-dark cycle.

*Cranial window preparation*. At the age of 4-week old, the mice were deeply anesthetized and then locked onto a stereotaxic platform. After prepping, a 4 mm-wide craniotomy was drilled over the motor cortex using a high-performance surgical drill. The center of the craniotomy was ~1.6 mm lateral to the bregma[37]. About 300 nL of AAV/DJ-flex-GCaMP6m virus was injected into the forelimb area of the right motor cortex (1.5 mm lateral and 0.3 mm anterior to the bregma and 300 μm in depth, according to previous studies[38,39]) via a 1.0 mm O.D. glass microneedle with a 10–20 μm diameter tip attached to a Nanoject microinjector pump (Nanoject II, Drummond). A 100 μm-thick glass coverslip was then placed over the exposed brain and sealed to the skull with vet glue. A 1-g titanium head restraining bar was attached to the head with "cold cure" denture material for later attachment of the miniature imaging probe. The transgene expression was checked 3 weeks after the initial surgery with a tabletop two-photon microscope.

*In vivo imaging*. After the transgene expression was successfully confirmed, the mice would be used for 2P fiberscopy imaging in vivo. For head-fixed imaging, we restrained the mouse by locking the head-restraining bar to a home-made platform and the 2P probe was gently placed against the cranial window surface. We used an

external 3-D translation stage to adjust the position of the probe for freely behaving imaging, the imaging probe was secured to the head restraining bar through a customized adaptor. After a suitable FOV was identified, the mouse was released and allowed to walk/behave freely within a home-built imaging platform (Fig. 4a). One camera (BFLY-PGE-12A2M-CS, FLIR) was set above the platform to obtain the top view of the freely behaving mouse in synchronization with 2P imaging.

**Data processing**. We used ImageJ and MATLAB® for image processing, including frame averaging and color mapping. We applied a non-rigid registration method (NoRMCorre[32]) to correct motion artifacts before processing the in vivo images and a well-established post-processing pipeline (CaImAn[23]) to recognize/segment neuron somas for dynamic signal analysis. It is noted that the motion correction was not performed or needed for DNN training and testing. It was only used when we analyzed the neuronal calcium signals. Here calcium signals were presented as the time-dependent GCaMP fluorescence intensity. It was calculated by averaging over the whole ROI for a given neuron soma. After background fluorescence subtraction, the dynamic fluorescence trace was normalized to the maximum value and expressed as the relative fluorescence change ($\Delta F/F$). The signal-to-noise ratio (SNR) for a given neuron was defined as:

$$\text{SNR} = 10\log_{10}\left(\frac{P_{\text{signal}}}{P_{\text{noise}}}\right), \tag{1}$$

where $P_{\text{signal}}$ was the average intensity for a given neuron and $P_{\text{noise}}$ was the average background noise adjacent to the neuron. To quantify the calcium signal fidelity after DNN processing, we calculated the normalized root mean square error (NRMSE) between the calcium signals of DNN output image and the input image (raw data) for a given active neuron, and the NRMSE was defined as:

$$\text{NRMSE}_{\text{neuron}} = \frac{\text{RMSE}_{\text{neuron}}}{\max(\text{F}) - \min(\text{F})} = \frac{\sqrt{\frac{\sum_{t=1}^{T}(F(t) - F_0(t))^2}{T}}}{\max(F) - \min(F)}, \tag{2}$$

where $F(t)$ is the normalized $\Delta F/F$ for a given neuron at time $t$ (i.e., the $t$th frame of the time-series imaging dataset) in the DNN output image, $F_0(t)$ is the corresponding input (raw data) $\Delta F/F$, and $T$ represents the total number of frames. The expression $\max(F) - \min(F)$ represents the maximum range of $\Delta F/F$ for the given neuron. To assess the performance of the DNNs for image quality enhancement, we calculated the multi-scale structure similarity (MS-SSIM) index[40] and the normalized root mean square error (NRMSE) between the DNN output images and the corresponding ground truth or reference images. Here the NRMSE for a given image frame was defined as:

$$\text{NRMSE}_{\text{image}} = \frac{\text{RMSE}_{\text{image}}}{\max(\text{I}) - \min(\text{I})} = \frac{\sqrt{\frac{\sum_{j=1}^{m}\sum_{k=1}^{n}(I(j,k) - I_0(j,k))^2}{mn}}}{\max(I) - \min(I)}, \tag{3}$$

where $I(j, k)$ is the intensity at the position $(j, k)$ in a given DNN-enhanced image and $I_0(j, k)$ is the intensity at the position $(j, k)$ in the ground truth (or reference) image. The expression $\max(I) - \min(I)$ represents the range of intensity for the given DNN-enhanced image. Once the NRMSE of the entire testing set was calculated, their mean and standard deviation were computed and plotted in Fig. 3e.

**Deep neural network (DNN)**. In this work, we adopted a deep neural network (DNN) based on conditional generative adversarial network (cGAN)[19,20]. In this framework, two sub-networks were used simultaneously: a generative network learned how to enhance the SNR for 8-bit $512 \times 512$ 2D monochrome images and a discriminative network returned an adversarial loss between the enhanced image and the corresponding ground truth[20]. Specifically, we aimed to optimize the following objective function:

$$\min_G \max_D \mathcal{L}_{cGAN} = E_{x,y}[\log D(x, y)] + E_{y,z}[\log(1 - D(G(z, y), y)]. \tag{4}$$

Here $x$ represents the ground truth, $y$ represents the input images (e.g., high frame rate images), and $z$ represents a random noise vector that works together with the input images to generate output images $G(z, y)$. The discriminator $D(*)$ takes an input that can be either a ground truth image $x$ or a "fake" image "generated" by the generator $G(z, y)$ and returns the probability of the input to be "true" with a conditional input of $y$. $E_{x,y}[*]$ represents the mean value of $\log D(x, y)$ over the entire training dataset (i.e., the ground truth and the input images), and $E_{y,z}[*]$ represents mean value of $\log(1 - D(G(z, y), y)$ over all the "fake" images generated by $G(z, y)$. In the generator design, we adapted a U-Net structure to improve the imaging resolution for the input images based on the ground truth[41]. In the discriminator design, we restricted both fine and coarse spatial structural information by combining an L1 norm (least absolute deviations) and a convolution neural network (CNN)-based PatchGAN structure[41]. The combining weights and the patch scale were tuned as a hyperparameter[42]. We followed the standard procedure to update the parameters for the DNN[43] and adopting the Adam solver during optimization[44]. To test the training stability, we conducted five separate training cycles for DNN-1 and DNN-2 with different initializations, and the results confirmed that the training was stable (see Supplementary Table 2 and Supplementary Fig. 10 for details).

The program was implemented using Python v3.6, and the DNN was implemented using Pycharm (2020.2.3) and torch (0.4.1). We used a PC with an Intel Core i7-8700K CPU 3.70 GHz (6 cores), 32 GB system RAM, and an NVIDIA GPU (GeForce RTX 2080 Ti, 11 GB RAM), running a Microsoft Windows 10 professional operating system. Other data processing was performed by our customized codes, which were implemented using MATLAB® (R2020a, Mathworks). It took about 205 min to train each DNN model (with a total of 500 frames for both DNN-1 and DNN-2, $512 \times 512$ pixels/frame, 8 bits/pixel, and a total of 200 training epochs). Motion correction took about 5 min for 2000 frames ($512 \times 512$ pixels/frame, 8 bits/pixel) on the same computer platform.

**Reporting summary**. Further information on research design is available in the Nature Research Reporting Summary linked to this article.

## Data availability
All data used and reported in this study have been deposited in the Figshare (https://doi.org/10.6084/m9.figshare.19193792).

## Code availability
The deep-learning platform used in this study was adapted from a publicly available repository: https://github.com/junyanz/pytorch-CycleGAN-and-pix2pix. We recommend interested readers use the most updated pix2pix routine for compatibility concern. Our customized source codes are also available upon request by sending an email to jhu.bme.bit@gmail.com.

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

## Acknowledgements

The authors would like to acknowledge the funding supports from the National Institutes of Health (NIH) (R01 CA153023, Li), the National Science Foundation (NSF) Major Research Instrumentation (MRI) grant (CEBT1430030, Li), the Johns Hopkins Medicine Discovery Fund Synergy Award (Bergles and Li), and the Bisciotti Foundation through Johns Hopkins Technology Ventures (Li and Park).

## Author contributions

D.L. and H.G. conceived the method. H.G. and A.L. performed the imaging experiments and H.G. carried out the data analyses. D.L. developed the neural network framework. H.P. developed the high-speed fiberscope. Y.Y. prepared the animal model for in vivo experiments. Y.T.A.G. prepared brain slices for ex vivo experiments under the supervision of D.E.B. M.L. contributed to the development of the fiber-optics used in the fiberscope. H.G. and D.L. drafted the manuscript. All authors provided critical feedback to the manuscript. X.L. proposed the intermediate DNN training and validation scheme, supervised the overall project design and execution, and led the effort in manuscript editing.

## Competing interests

The authors declare no competing interests.
