## [Peer Review File · Nature Communications]

Reviewers' Comments:

Reviewer #1:

Remarks to the Author:

Herein, the authors develop a deep learning framework as an attempt to solve the speed bottleneck inherent to endomicroscopic imaging of the mouse brain *in vivo*. The two-step DL routine presented utilizes “transfer learning” to properly train the authors’ model for high fidelity image reconstruction at speeds reaching 26 fps. Together with the combination of a novel imaging stratagem, the reviewer has no doubt that the work detailed herein is of great interest to researchers across many scientific disciplines (DL, neuroscience, fundamental biology, etc.). However, as the authors likely know, the field of deep learning has been inundated with work applied to biological imaging that is (understandably) challenging for non-practitioners to trust and for others to reproduce as of late. With this said, it would be a great disservice to the community not to look at the methodology and results presented herein through a particularly critical lens.

I will list my concerns and comments below – but first I would like to ensure that I have read the core details about the presented work correctly.

From the information provided...

- It seems that the authors have used a GAN (modeled after pix2pix), along with 2P-endomicroscopy of mouse brain tissue *ex vivo* to train a model to act as a denoiser. Afterwards, the authors used this successfully trained GAN (referred to as “DNN-1”) along with images acquired using 2P-endomicroscopy *in vivo* (mouse kept stable during imaging) to generate denoised images at high scan rates. These denoised *in vivo* images were then used to train a second GAN (with equivalent network architecture, both generator and discriminator, to that of DNN-1, referred to herein at “DNN-2”). Further, the weights from DNN-1’s generator (and perhaps discriminator?) were kept upon initial training of DNN-2 (implied by use of “transfer learning”). With this, DNN-2 was encouraged to focus less on the denoising aspect of the inverse mapping *per se* (already learned in step #1) and more so on correctly encoding valuable information present *in vivo* during the training phase. After satisfactory convergence, DNN-2’s trained generator was used to both denoise and correctly encode *in vivo* images acquired during motion at different scan rates (an example of which illustrated in Supplementary Video #3).

^ If the reviewer does not have this story quite right, please correct where needed.

Concerns/Questions:

- Let us say that another group would like to use this technique for their 2P-endomicroscope. How amendable would this analytic routine be to that groups’ protocol?

- From the reviewer’s perspective, it seems that the *in vivo* data used for training DNN-2 was collected using the exact same FOV in the exact same mouse subject as the subject’s movement was restrained. Then, the test data seems to be the motion-corrected 2P-endomicroscopy data acquired after allowing the subject to move freely. This said, could the network herein be used

for another mouse subject? If so, how challenging would it be to ensure images similar enough to that used in DNN-2's training pipeline on this separate imaging experiment? Or would DNN-1 (denoiser trained on mouse brain data *ex vivo*) need to be trained *again* with data representative of new FOV's if the depth of field was slightly altered, the excitation wavelength used was different, etc.?

- Does a separate network need to be trained for the target scan rate, or does DNN-2 generalize to more than one scan rate?

- Time. How long does it take to train each model? How long does motion-correction using the non-rigid registration algorithm of choice take?

- How stable is training? To address training stability, the reviewer recommends including loss curves over five separate training cycles (at minimum) and a set number of epochs (different initializations for DNN-1, equivalent weights for DNN-2 initialization) in a supplementary document. The authors can display this using the average and standard deviation along with using a toolbox such as the following:

(*MATLAB*): <https://www.mathworks.com/matlabcentral/fileexchange/26311-raacampbell-shadederrorbar>

- *“For head-fixed imaging, we restrained the mouse by locking the head-restraining bar to a home-made platform and the two-photon probe was gently placed against the cranial window surface. We used an external 3-D translation stage to adjust the position of the probe for freely-behaving imaging, the imaging probe was secured to the head restraining bar through a customized adaptor. After a suitable FOV was identified, the mouse was released and allowed to walk/behave freely within a home-built imaging platform (Fig. 4a). One CCD camera (BFLY-PGE-12A2M-CS, FLIR) was set above the platform to obtain the top view of the freely-behaving mouse in synchronization with two-photon imaging.”*

From what the reviewer has gathered, the data collected for training DNN-1 was mostly for the purpose of acting as a denoiser. This said, the reviewer is curious as to whether the data collected for this purpose must be of only mouse brain? For instance, in the case of the well-known AUTOMAP [1], authors used the freely available ImageNet to generate the data for their workflow – a workflow which was subsequently capable of end-to-end inverse mapping using four different encoding schemes without the training dataset including any brain images.

The reviewer is not suggesting that the authors herein explain why they didn't use a simulation data routine. The reviewer is suggesting that, if the argument claiming “DNN-1 is used as *purely* a denoiser” holds, then an equivalent network trained to map randomly acquired data collected through the 2P-endomicroscope to fully-sampled ground-truth would act as both: **1)** a denoising network which performs equivalent inverse mapping and **2)** a MORE robust denoiser given that it would be trained in a fashion much further removed from the application of interest compared to that described by the authors.

Indeed, (apologies for referring to this work again, but it is a very clear example) the developers of AUTOMAP did not use a GAN-like framework – but a relatively simplistic MLP -> CNN. To the reviewer, this sort of DL-based routine is more forgiving **strictly as a denoiser** given that GANs have a few notable drawbacks (repeatability is not user-friendly, ultra-sensitive to hyperparameter settings, WILL “overfit” if not trained properly [**Example #1** provided for simplicity], etc.).

Example #1. Pix2pix trained to encode simplistic illustrations to cats applied to both sensical data input (A) and nonsensical input (B).

Given that the authors chose a cGAN framework in contrast to a more computationally friendly, strictly one-to-one mapping routine for image denoising, the reviewer believes that DNN-1 herein acts as *significantly* more than just an “image denoiser” and is heavily biased towards assisting the authors *very* specific application. Since the authors use the output of DNN-1 on the *in vivo* data, as well as DNN-1’s weights for training DNN-2, it is hard to believe that the results were not artificially improved by a significant margin due to intrinsic bias learned by the denoiser. Given the authors’ resources, a relatively quick experiment to prove this is not so (using the 2P-endomicroscope to collect a variety of image data in the same way as detailed herein for training a denoiser) should be relatively easy to undertake and would, in this reviewer’s humble opinion, **greatly** validate the robustness of the work herein.

Additionally, the reviewer believes that training DNN-1’s generator (which, unless the reviewer is mistaken, is the *exact* same generator architecture as DNN-2) in a strictly end-to-end fashion and subsequently using this “generator” for both denoising and initializing the weights of DNN-2’s generator would *also* offer higher generalizability/robustness. Moreover, if the authors train a denoiser in this fashion to map 2P-endomicroscopy data acquired using a dataset which is mostly comprised of experimentally acquired data *other* than mouse brain data, and use *this* denoising network (*non-GAN*) for their routine, this would end all doubt the reviewer currently has regarding intrinsic bias and would significantly enhance the quality/novelty of the work presented herein.

- “We applied a non-rigid registration method (**NoRMCorre** [37]) to correct motion artifacts before processing the *in vivo* images...”

As the authors likely agree, the workflow presented herein would be slightly more intriguing had the authors been able to use their workflow for denoising, encoding and motion artifact correction simultaneously. There are many places herein where not enough emphasis is placed

on the fact that motion-artifact correction is NOT performed by DL, but by a preprocessing step. The reviewer suggests that this important point is pointed out more often and emphasized more strongly where needed – especially since the training stratagem described would definitely not lend itself to a DNN-2 capable of behaving properly when used for data heavily corrupted by motion-related artifacts. For example, the caption of Supp. Visualization #3 – a visualization which will likely become very popular and thus makes the need for proper emphasis even more crucial.

- *“We conducted video-rate (26 fps) two-photon imaging in freely-behaving mice, demonstrating the applicability of this DNN-assisted two-photon endomicroscope for high-speed neural functional imaging.”*

As the above point addressed, without motion-correction 2P-endomicroscopy wouldn't have been possible in *“freely-behaving mice”*. Therefore, this is one example where the reviewer recommends the wording be changed to something along the lines of *“... DL routine coupled with conventional motion correction for high-speed neural functional imaging via a two-photon endomicroscope.”*

- *“For image denoising, many classical digital image processing methods are available, such as non-local mean filter [25], anisotropic filtering [26], and wavelet-based methods [27]. However, these methods require sophisticated mathematic modeling, parameter tuning and are often based on mathematical priors that may not be universally applicable.”*

The reviewer understands the negatives behind the three techniques listed. Though, given that the denoiser herein is used to generate ground-truth labelling for DNN-2, there is no reason why a denoising performance assessment cannot be undertaken using at least one of the listed techniques. The reviewer also understands that the methods do indeed require “parameter tuning” – though GANs also require hyperparameter tuning and arguably even *greater* practitioner knowledge given the high potential for misuse (especially for an application as important as this), relative decrease in computational friendliness, etc.

Thus, the reviewer suggests that the authors include supplementary results comparing the results of the conventional denoiser of choice versus that of DNN-1 (which again, the reviewer strongly believes is biased as currently presented). Further, to ensure that the conventional denoiser is used *properly*, the reviewer recommends against the use of default parametric settings. An example of this for TVAL can be seen in **Figure 6** of [2].

Other remarks:

In closing, the reviewer believes that this work has potential for being greatly suitable for Nature Communications, as well as being a seminal contribution to science, with just a little improvement. Altogether, the reviewer greatly enjoyed reading this manuscript and commend the authors on their development of a novel DL-based technique to improve the capability of a remarkable imaging methodology.

References

[1] Zhu, Bo, et al. "*Image reconstruction by domain-transform manifold learning.*" *Nature* 555.7697 (2018): 487-492.

[2] Ochoa, M., et al. "*High compression deep learning based single-pixel hyperspectral macroscopic fluorescence lifetime imaging in vivo.*" *Biomedical Optics Express* 11.10 (2020): 5401-5424.

Reviewer #2:

Remarks to the Author:

In this manuscript, the authors have developed a two-photon microendoscopy imaging method where data is recorded at low speeds, limited by the actuator's ability to steer the laser beam from place to place, and extrapolated to high-resolution and high speed using deep learning inference.

My first major comment is that the manuscript in its current format doesn't adequately clarify what the new contributions to the methodology are. The authors should clarify where the novelty is. While the abstract says "A High-speed scanner [...] were developed", the technology described in the methods does not seem to be any different from previous hardware designs that the authors already published. In fact, the authors clarify in the conclusion that "A faster scanner, combined with our deep learning-based approach, may increase the frame rate beyond that achieved here", indicating that no new hardware was developed. The remaining novelty of the manuscript is in the data acquisition parameters, and in the algorithms for data processing. This point should be clarified to properly identify new work vs. previously published work.

The algorithm introduced by the authors is a deep learning method trained offline to infer a high speed-high resolution output from raw data sampled sparsely. The DNN1 trained on high-resolution ex-vivo data performs denoising of raw images by trying to mimic averaging. DNN2 takes low-resolution and low SNR images to predict high-resolution and high SNR images. DNN2 performs both denoising and resolution enhancement tasks. In order to train DNN2, the authors used DNN1 to create ground truth data for the output of DNN2 and the input of DNN1 is the downsampled version of the output of DNN1.

The proposed approach to obtain the training data is innovative, however, the utilization of DNNs to denoise and upsample imaging data is well established in the computer vision community. Having seen several research papers utilize those methods for quantitative scientific imaging, caution should be taken to adequately interpret the meaning of the output images.

For instance, the authors claim that the algorithm enables higher resolution and higher imaging speed. Yet, the output of a CNN is an inference of what could be the most likely a high-resolution output, hence, the image metrics (e.g. SNR, minimal feature size) depend on the typical value of those parameters in the training data rather than on what is supplied as input. In other words, the claims of resolution capabilities, often relate to the apparent resolution of output images, rather than the ability to resolve - distinguish - features in the sample.

In this manuscript, the data being presented shows calcium transient events in labeled neurons. In all cases where calcium activity is strong and occupies the entire cell soma, estimated dF/F signals obtained by analysis of raw or enhanced data are about the same, and those events could have been detected without CNN enhancement, denoising does not unveil additional information.

My review aimed to identify what could be the benefits of the proposed image enhancement.

In the absence of simultaneously recorded high-resolution images, it is unclear if the dendrites identified in Figure 2d genuinely correspond to dendrites physically present at the depicted locations, or if they are only an interpretative display that best estimates where hypothetical dendrites should be placed to yield the acquired low resolution image. Similarly, the brief events that occur in speed enhanced outputs, are representations that may not be accurate. In other words, processing raw data through a DNN does not create new information, but reorganizes it in an output that appears to be of superior quality to the human eye. The proposed approach may have benefits, but only if the authors can make a case that enhancing the display with their algorithm enables the detection of features or events that are 1-verifiable with ground truth and 2

- not already visible in the raw data.

The manuscript in its current format should undergo major revisions, first to adequately classify novelty vs previously published work, then to better show the benefits of the proposed algorithm. The intrinsic resolution of the output images should be distinguished from the imaging resolution that characterizes the ability of the device to detect / separate physical features.

Minor comments

A scale bar would be most welcome in the supplementary videos to recall the dimensions of the field of view.

Detailed point-by-point responses:

Authors' responses are in blue, changes made to the manuscript in yellow, and all line numbers refer to the submitted documents.

Reviewers' Comments:

Reviewer 1

General comments:

Herein, the authors develop a deep learning framework as an attempt to solve the speed bottleneck inherent to endomicroscopic imaging of the mouse brain in vivo. The two-step DL routine presented utilizes “transfer learning” to properly train the authors' model for high fidelity image reconstruction at speeds reaching 26 fps. Together with the combination of a novel imaging stratagem, the reviewer has no doubt that the work detailed herein is of great interest to researchers across many scientific disciplines (DL, neuroscience, fundamental biology, etc.). However, as the authors likely know, the field of deep learning has been inundated with work applied to biological imaging that is (understandably) challenging for non-practitioners to trust and for others to reproduce as of late. With this said, it would be a great disservice to the community not to look at the methodology and results presented herein through a particularly critical lens. I will list my concerns and comments below – but first I would like to ensure that I have read the core details about the presented work correctly.

Response:

We very much appreciate the reviewer's positive general comment on our work and its potential impact. We would also like to thank the reviewer for his/her constructive suggestions.

Comment 1:

It seems that the authors have used a GAN (modeled after pix2pix), along with 2P endomicroscopy of mouse brain tissue ex vivo to train a model to act as a denoiser. Afterwards, the authors used this successfully trained GAN (referred to as “DNN-1”) along with images acquired using 2P endomicroscopy in vivo (mouse kept stable during imaging) to generate denoised images at high scan rates. These denoised in vivo images were then used to train a second GAN (with equivalent network architecture, both generator and discriminator, to that of DNN-1, referred to herein at “DNN-2”). Further, the weights from DNN-1's generator (and perhaps discriminator?) were kept upon initial training of DNN-2 (implied by use of “transfer learning”). With this, DNN-2 was encouraged to focus less on the denoising aspect of the inverse mapping per se (already learned in step #1) and more so on correctly encoding valuable information present in vivo during the training phase. After satisfactory convergence, DNN-2's trained generator was used to both denoise and correctly encode in vivo images acquired during motion at different scan rates (an example of which illustrated in Supplementary Video #3). ^ If the reviewer does not have this story quite right, please correct where needed.

Response:

Thank the reviewer for the excellent summary. Most of it is accurate. There are only a few descriptions that were not exactly what we meant which might be caused by our unclear presentation in the previous submission. We would like to take this opportunity to clarify those points:

(1) Regarding “Further, the weights from DNN-1's generator (and perhaps discriminator?) were kept upon initial training of DNN-2 (implied by use of “transfer learning”)”.

Response: This is an excellent comment and we apologize for not making it clear in our previous manuscript. In our original training strategy, we **did not** transfer the weights from DNN-1’s generator or discriminator to DNN-2.

Inspired by the reviewer’s comment, we retrained DNN-2 and compared the training performance (using the loss curves) under the following two conditions: 1) Training the network from scratch (initializing the generator and discriminator of DNN-2 with random weights). That was what we did in the original manuscript (denoted as “Scratch”); 2) Training the network using a pre-trained model (initializing the generator and discriminator of DNN-2 with weights inherited from the trained DNN-1, denoted as “Pretrain”). We chose $M=2, 4, 8,$ and 32 as examples, and the loss curves are shown in the following figure.

According to the above loss curves, the “Pretrain” training method had faster convergence at the beginning (0-40 epochs). Afterwards, the performances of the two training methods were similar. The loss curves turned to be nearly identical at the end (160-200 epochs). The results have been added to the revised **Supplementary Information** (see **Figure S3**).

(2) Regarding “DNN-2 was encouraged to focus less on the denoising aspect of the inverse mapping per se (already learned in step #1) and more so on correctly encoding valuable information present *in vivo* during the training phase.”

Response: This is an excellent comment and we apologize for not making it clear in our previous manuscript. The DNN-2 was trained to achieve both functions simultaneously: (1) image denoising and (2) pixel inpainting (up-sampling), considering the input training data to DNN-2 were *in vivo* noisy images of a low sampling density and the ground truth data were denoised images of a high sampling density.

The manuscript has been revised accordingly in response to the above comments. See Page 11, Lines 201-203 in the main manuscript (texts highlighted in yellow):

“

In addition, the DNN-2 inherited the function of SNR improvement from DNN-1; thus DNN-2 was trained to achieve 1) image denoising and 2) pixel inpainting (up-sampling) simultaneously.

”

And Page 12, Lines 228-235:

“

In addition, we introduced another protocol to train the DNN-2. We initialed the generator and discriminator of DNN-2 with the pre-trained weights (inherited from DNN-1, denoted as “Pretrain”) rather than random weights (denoted as “Scratch” which was used in the above DNN-2 training). Compared with “Scratch” scheme, the “Pretrain” configuration had faster convergence at the beginning. The two training methods were similar to each other after a certain number of epochs (~40) and the loss curves became nearly identical (see **Supplementary Information Figure S3** for more details). If the training dataset gets significantly larger and the computation cost becomes the major constraint, the “Pretrain” scheme will be an effective alternative.

”

Comment 2:

Let us say that another group would like to use this technique for their 2P-endomicroscope. How amendable would this analytic routine be to that groups’ protocol?

Response:

We would like to thank the reviewer for this excellent question. Considering the system-dependent data for the two-stage training procedure are only images collected with the same system (under different conditions), it is expected that similar training processes can be used for other similar 2P-endomicroscopy systems. Due to potentially different noise features, the two DNNs are expected to be re-trained with the relevant data collected by the given system.

Comment 3:

From the reviewer’s perspective, it seems that the *in vivo* data used for training DNN-2 was collected using the exact same FOV in the exact same mouse subject as the subject’s movement was restrained. Then, the test data seems to be the motion-corrected 2P-endomicroscopy data acquired after allowing the subject to move freely. This said, could the network herein be used for another mouse subject? If so, how challenging would it be to ensure images similar enough to that used in DNN-2’s training pipeline on this separate imaging experiment? Or would DNN-1 (denoiser trained on mouse brain data *ex vivo*) need to be trained again with data representative of new FOV’s if the depth of field was slightly altered, the excitation wavelength used was different, etc.?

Response:

This is a great, very insightful question/comment about the overall training flow. We apologize that we did not make the relevant key point clear in our previous manuscript. The mice used for collecting *in vivo* data for training DNN-2 were **excluded** from those we used to perform freely-moving imaging. Thus, the training images for DNN-2 were independent from the *in vivo* images to which the trained DNN-2 was later applied, and the imaging FOVs would be different.

Regarding the training routine for DNN-1, the *ex vivo* data collected from brain slices were totally distinct from *in vivo* images. To better prepare DNN-1 for use on the *in vivo* two-photon images (i.e., for generating the ground truth for DNN-2 training), we intentionally diversified the DNN-1 training data by

collecting images from many (about 100) different FOVs and also added random white noise to the training data. Similarly, we prepared the DNN-2 training data set by collecting images from different FOVs on multiple mice.

Therefore, if the FOV or the depth of field is altered when collecting the DNN-2 input data, DNN-1 does not need to be retrained. If the change of excitation wavelength is large enough to impact the noise performance of the system, DNN-1 has to be retrained.

We have added more explicit descriptions in the revised manuscript related to the above discussion. See Page 11, Lines 210-211:

“

The corresponding testing set II (Fig. 3a, acquired under the same conditions as training set II but from different mice)

”

and Page 14, Lines 249-250:

“

The mice used to perform freely-moving imaging were different from those head-fixed ones used for collecting the *in vivo* data for training DNN-2.

”

Comment 4:

Does a separate network need to be trained for the target scan rate, or does DNN-2 generalize to more than one scan rate?

Response:

We thank the reviewer for this very valuable question. One DNN-2 corresponds to one scan rate. In the manuscript, we used the M value to represent the down-sampling ratio (which is proportional to the imaging frame rate).

To make it much clearer, we revised the manuscript accordingly (see Pages 6-7, Lines 116-117):

“

It is noted that a given DNN-2 corresponds to one scanning density. We need to train the DNN-2 separately for images collected with a different scanning density.

”

and Page 10, Line 188:

“

Each scanning density requires a separately trained network.

”

The similar approach is introduced when dealing with other brain imaging modalities such as MRI, one example is SMORE[1], where different down-sampling factors require separate training.

Comment 5:

Time. How long does it take to train each model? How long does motion-correction using the non-rigid registration algorithm of choice take?

Response:

We appreciate the reviewer’s valuable questions. The time needed for training each model (both DNN-1 and DNN-2) was about 205 minutes with a total training dataset size of 500 frames, 512 x 512 pixels per frame, and 8 bits per pixel. The key hardware parameters of our computer platform are listed below:

Operating System	Windows 10 64-bit
Processor	Intel Core i7-8700K CPU @ 3.70GHz (6 cores)
System RAM	32 GB
GPU	NVIDIA GeForce RTX 2080 Ti
GPU RAM	11 GB

Motion correction took about 5min for 2000 frames (512 x 512 pixels per frame, 8 bits per pixel) on the same computer platform.

We have revised manuscript accordingly (see Pages 23-24, Lines 448-454):

“

All the data analyses, DNN training and applications were carried out in the Pycharm (2020.2.3) and MATLAB (R2020a) environment running on a PC with Intel Core i7-8700K CPU 3.70GHz (6 cores), 32 GB system RAM, and an NVIDIA GPU (GeForce RTX 2080 Ti, 11 GB RAM). It took about 205 minutes to train each DNN model (with a total of 500 frames for both DNN-1 and DNN-2, 512 x 512 pixels/frame, 8 bits/pixel, and a total of 200 training epochs). Motion correction took about 5 minutes for 2000 frames (512 x 512 pixels/frame, 8 bits/pixel) on the same computer platform.

”

Comment 6:

How stable is training? To address training stability, the reviewer recommends including loss curves over five separate training cycles (at minimum) and a set number of epochs (different initializations for DNN-1, equivalent weights for DNN-2 initialization) in a supplementary document. The authors can display this using the average and standard deviation along with using a toolbox such as the following:

(*MATLAB*): <https://www.mathworks.com/matlabcentral/fileexchange/26311-raacampbellshadederrorbar>

Response:

We would like to thank the reviewer for this extremely valuable comment. Following the reviewer’s suggestion, we trained DNN-1 and DNN-2 (M=8, “Scratch” training scheme) with five different initializations to test the stability and the loss curves are illustrated in the following figure.

The loss curves show that the training was stable against distinct initializations. After 150 epochs, the differences became smaller and the loss curves converged to the same value for different initializations. As for DNN-1, the relative standard deviations (RSDs, defined as the standard deviation divided by the average over five tries) of the G loss and D loss at the endpoint were 3.4% and 9.3%, respectively. As for DNN-2, the RSDs of the G loss and D loss at the endpoint were 6.0% and 10.7%, respectively. The results have been added to the revised **Supplementary Information** (see **Table S2** and **Figure S7**).

We also revised the manuscript accordingly (see Page 23, Lines 444-457):

“

To test the training stability, we conducted five separate training cycles for DNN-1 and DNN-2 with different initializations, and the results confirmed that the training was stable (see **Supplementary Information Table S2** and **Figure S7** for details).

”

Comment 7:

From what the reviewer has gathered, the data collected for training DNN-1 was mostly for the purpose of acting as a denoiser. This said, the reviewer is curious as to whether the data collected for this purpose must be of only mouse brain? For instance, in the case of the well-known AUTOMAP [1], authors used the freely available ImageNet to generate the data for their workflow – a workflow which was subsequently capable of end-to-end inverse mapping using four different encoding schemes without the training dataset including any brain images.

The reviewer is not suggesting that the authors herein explain why they didn't use a simulation data routine. The reviewer is suggesting that, if the argument claiming “DNN-1 is used as purely a denoiser” holds, then an equivalent network trained to map randomly acquired data collected through the 2P-endomicroscope to fully-sampled ground-truth would act as both: 1) a denoising network which performs equivalent inverse mapping and 2) a MORE robust denoiser given that it would be trained in a fashion much further removed from the application of interest compared to that described by the authors.

Response:

We very much appreciate this reviewer's excellent suggestion. The primary function of our customized DNN-1 is to restore “neurons” from noisy background. Besides denoising, DNN-1 also helps perform the end-to-end inverse mapping for specific image features, and in our case, the specific features are neurons including soma and associated axons and dendrites.

We have revised the manuscript to make this point clearly stated. See Pages 9-10, Lines 177-183:

“

It is worth mentioning that the primary purpose of DNN-1 was to restore “neuronal” features from noisy background. Besides denoising, DNN-1 also performed the end-to-end inverse mapping to preserve sharp edges and fine structural details rather than over-smooth textures. To achieve optimal image restoration, the training and testing dataset should share the same specific image features or cell types, which include soma, associated axons and dendrites in this work. Applying a trained DNN-1 to images of an unknown cell type would generate unwanted artifacts, and one example is presented in **Supplementary Information Figure S2**.

”

The DNN-1 we used in the manuscript was trained with “neuron” images, and it is denoted as DNN-1(a) in this case. If DNN-1 is trained by other dataset rather than “neuron” images, it will introduce unwanted features when the trained network is applied to “neuron” images. One example is shown below, where we trained the DNN-1 with *ex vivo* images acquired with the same two-photon endomicroscope but from another type of brain slices with GCaMP6s-expressing **astrocytes**, following the same training protocol as DNN-1(a). This new network is denoted as DNN-1(b). When we applied the trained DNN-1(a) and DNN-1(b) to “neuron” images collected from *ex vivo* GFP-immunostained mouse brain slices (noting the testing data was excluded from the training set of DNN-1(a)), the results demonstrate that the fine features of DNN-1(b) output are obviously distorted compared with the DNN-1(a) output and the ground truth. It is evident on the DNN-1(b) output that the soma is blurred (indicated by subpanel (b), ROI 1), and the dendrites (indicated by subpanel (b), ROI 2) are difficult to resolve.

The above training results are included in **Supplementary Information Figure S2**.

We would like to mention that the idea about AUTOMAP is of great interest to us. We plan to look into the details in the near future as a continuation of the current research.

Comment 8:

Given that the authors chose a cGAN framework in contrast to a more computationally friendly, strictly one-to-one mapping routine for image denoising, the reviewer believes that DNN-1 herein acts as significantly more than just an “image denoiser” and is heavily biased towards assisting the authors very specific application. Since the authors use the output of DNN-1 on the *in vivo* data, as well as DNN-1’s weights for training DNN-2, it is hard to believe that the results were not artificially improved by a significant margin due to intrinsic bias learned by the denoiser. Given the authors’ resources, a relatively quick experiment to prove this is not so (using the 2P-endomicroscope to collect a variety of image data in the same way as detailed herein for training a denoiser) should be relatively easy to undertake and would, in this reviewer’s humble opinion, **greatly** validate the robustness of the work herein(TBD).

Additionally, the reviewer believes that training DNN-1’s generator (which, unless the reviewer is mistaken, is the exact same generator architecture as DNN-2) in a strictly end-to-end fashion and subsequently using this “generator” for both denoising and initializing the weights of DNN- 2’s generator would also offer higher generalizability/robustness. Moreover, if the authors train a denoiser in this fashion to map 2P-endomicroscopy data acquired using a dataset which is mostly comprised of experimentally acquired data other than mouse brain data, and use this denoising network (non-GAN) for their routine, this would end all doubt the reviewer currently has regarding intrinsic bias and would significantly enhance the quality/novelty of the work presented herein.

Response:

The reviewer is correct about the functions of DNN-1. As discussed above, DNN-1 is primarily for denoising the neuron images, but it also helps restore the sampling density (or imaging resolution) of the neuron images. Due to the challenges in obtaining good quality neuronal images *in vivo* from freely-moving mice, we introduced DNN-1 as a bridge to construct the ground truth needed for training DNN-2, and the ultimate goal of DNN-2 is to compensate the loss in image quality caused by down-sampling and associated high frame-rate imaging. Indeed, it would be ideal to construct a network that is sufficiently accurate and broadly applicable to various images; but it is very challenging (as we discussed in the response to the previous comment). As an alternative, we chose a network (DNN-1) as the middle step for helping generate the dataset (ground truth) needed for training DNN-2.

Also as discussed before (see Response to Comment 1), in our strategy, DNN-2 did not inherit the weights from the trained DNN-1. The DNN-1 was only used to generate the ground truth for DNN-2. The DNN-2 will learn to achieve denoising and pixel inpainting (up-sampling) simultaneously.

Comment 9:

As the authors likely agree, the workflow presented herein would be slightly more intriguing had the authors been able to use their workflow for denoising, encoding and motion artifact correction simultaneously. There are many places herein where not enough emphasis is placed on the fact that motion-artifact correction is NOT performed by DL, but by a preprocessing step. The reviewer suggests that this important point is pointed out more often and emphasized more strongly where needed – especially since the training stratagem described would definitely not lend itself to a DNN-2 capable of behaving properly when used for data heavily corrupted by motion-related artifacts. For example, the caption of Supp. Visualization #3 – a visualization which will likely become very popular and thus makes the need for proper emphasis even more crucial.

As the above point addressed, without motion-correction 2P-endomicroscopy wouldn't have been possible in "freely-behaving mice". Therefore, this is one example where the reviewer recommends the wording be changed to something along the lines of "... DL routine coupled with conventional motion correction for high-speed neural functional imaging via a two-photon endomicroscope."

Response:

We thank reviewer for his/her very constructive comment on the motion-correction. We have revised the manuscript accordingly. See Page 21, Lines 396-398:

“

It is noted that the motion correction was not performed or needed for DNN training and testing. It was only used when we analyzed the neuronal calcium signals.

”

The motion-correction process was only employed when analyzing the neuronal calcium signals ($\Delta F/F$) from freely-moving imaging data. It is a necessary step to ensure credible neuron segmentation. Otherwise, the shift of field of view (induced by movement) would introduce fake calcium signals at apparently different locations. One key motivation to increase the imaging frame rate was to reduce motion artifacts.

With the help of DL-based solution, the output images show higher SNR and higher imaging resolution. The image quality improvements are beneficial for structure identification/segmentation, which are useful for motion-correction.

Comment 10:

The reviewer understands the negatives behind the three techniques listed. Though, given that the denoiser herein is used to generate ground-truth labelling for DNN-2, there is no reason why a denoising performance assessment cannot be undertaken using at least one of the listed techniques. The reviewer also understands that the methods do indeed require "parameter tuning" – though GANs also require hyperparameter tuning and arguably even greater practitioner knowledge given the high potential for misuse (especially for an application as important as this), relative decrease in computational friendliness, etc.

Thus, the reviewer suggests that the authors include supplementary results comparing the results of the conventional denoiser of choice versus that of DNN-1 (which again, the reviewer strongly believes is biased as currently presented). Further, to ensure that the conventional denoiser is used properly, the reviewer recommends against the use of default parametric settings. An example of this for TVAL can be seen in Figure 6 of [2].

Response:

We very much appreciate the reviewer's critique and recommendation on the comparison of different denoisers. Following reviewer's suggestion, we investigated several traditional denoisers and compared their performance with DNN-1 (the one described in the manuscript). The results show that DNN-1 offers greater global improvement (e.g., with a higher peak signal-to-noise ratio (PSNR) and multi-scale structural similarity index measure (MS-SSIM) relative to the ground truth). In addition, the DNN-1 outputs show better details (e.g., a clearer profile of cell bodies) than the traditional methods (the outputs of which often exhibit edge distortion/blur[2, 3]). The representative results are shown in the following figure:

We also evaluated the output image quality over different methods in terms of peak signal-to-noise ratio (PSNR) and multi-scale structural similarity index measure (MS-SSIM), the measurement results are shown below:

Method	PSNR (dB)	MS-SSIM
Original	25.43	0.82
NLM[1]	28.23	0.88
PURE-LET[2]	26.38	0.89
CANDLE[3]	28.43	0.90
DNN-1	30.99	0.92

We have added the results to the **Supplementary Information** (see **Figure S4** and **Table S1**). We have also revised the discussion about traditional denoisers in comparison with DNN-1. See Pages 15-16, Lines 286-292:

“

Usually, these methods require prior knowledge about the noise model of the images and a rational estimate about the noise level. In comparison, deep learning-based methods are advantageous. DNN can effectively figure out the system noise distribution and serve as a highly customized denoiser without the need for complex analyses of the noise model. Therefore, DNN shows better performance, especially when processing some fine structures. One example of qualitative and quantitative comparisons between some traditional imaging denoising methods and our reported DNN-1 is shown in **Supplementary Information Figure S4 and Table S1**.

”

Comment 11:

In closing, the reviewer believes that this work has potential for being greatly suitable for Nature Communications, as well as being a seminal contribution to science, with just a little improvement. Altogether, the reviewer greatly enjoyed reading this manuscript and commend the authors on their development of a novel DL-based technique to improve the capability of a remarkable imaging methodology.

Response:

We would like thank this reviewer again for his/her highly positive and encouraging comments on the potential value of this manuscript.

Reviewer 2

General comment 1:

In this manuscript, the authors have developed a two-photon microendoscopy imaging method where data is recorded at low speeds, limited by the actuator's ability to steer the laser beam from place to place, and extrapolated to high-resolution and high speed using deep learning inference.

My first major comment is that the manuscript in its current format doesn't adequately clarify what the new contributions to the methodology are. The authors should clarify where the novelty is. While the abstract says "A High-speed scanner [...] were developed", the technology described in the methods does not seem to be any different from previous hardware designs that the authors already published. In fact, the authors clarify in the conclusion that "A faster scanner, combined with our deep learning-based approach, may increase the frame rate beyond that achieved here", indicating that no new hardware was developed. The remaining novelty of the manuscript is in the data acquisition parameters, and in the algorithms for data processing. This point should be clarified to properly identify new work vs. previously published work.

Response:

We very much appreciate the reviewer's constructive comments.

The key innovation reported in this manuscript is the development and demonstration of a two-stage deep-learning strategy which involves transfer learning (e.g., using the trained DNN-1 to generate proper and otherwise unachievable ground truth for training DNN-2). This innovation enables improving imaging frame rate by more than 10-fold, making it possible to perform vide-rate (26 frames/second) 2P imaging in freely-moving mice with high image quality (i.e., with excellent imaging resolution and SNR that were previously not possible). The 2P endomicroscope hardware design was indeed similar to our previous ones except that the fiber-scanner adopted our most recent design which doubled the spiral scanning speed (from ~1.65kHz to 3.36kHz) through an optimized engineering protocol (and the engineering details can be found in reference [4] which are not the focus of this manuscript). We would also like to mention that this two-stage DNN-based method can be potentially applicable and valuable to other similar imaging modalities for increasing the frame rate while avoiding image quality loss. We have revised the manuscript accordingly to clarify the key innovation.

See Page 4, Lines 73-76:

“

This innovation enables 10-fold imaging frame-rate enhancement of endomicroscopy, making it feasible to perform vide-rate (26 fps) two-photon imaging in freely-moving mice with excellent imaging resolution and SNR that were previously not possible.

”

And Page 16, Lines 308-310:

“

This two-stage DNN-based method can be potentially applicable and valuable to other similar imaging modalities for increasing the frame rate while avoiding image quality loss.

”

General comment 2:

The algorithm introduced by the authors is a deep learning method trained offline to infer a high speed-high resolution output from raw data sampled sparsely. The DNN1 trained on high-resolution ex-vivo

data performs denoising of raw images by trying to mimic averaging. DNN2 takes low-resolution and low SNR images to predict high-resolution and high SNR images. DNN2 performs both denoising and resolution enhancement tasks. In order to train DNN2, the authors used DNN1 to create ground truth data for the output of DNN2 and the input of DNN1 is the downsampled version of the output of DNN1.

The proposed approach to obtain the training data is innovative, however, the utilization of DNNs to denoise and upsample imaging data is well established in the computer vision community. Having seen several research papers utilize those methods for quantitative scientific imaging, caution should be taken to adequately interpret the meaning of the output images.

Response:

We very much appreciate the reviewer's comment on the innovation. We also very much value the reviewer's caution on adequate interpretation of the images processed by DNNs. In Response to the later specific Comment 2, we also present some examples to further validate the method and demonstrate its benefit in restoring fine image features and enhancing the SNR.

Comment 1:

For instance, the authors claim that the algorithm enables higher resolution and higher imaging speed. Yet, the output of a CNN is an inference of what could be the most likely a high-resolution output, hence, the image metrics (e.g. SNR, minimal feature size) depend on the typical value of those parameters in the training data rather than on what is supplied as input. In other words, the claims of resolution capabilities, often relate to the apparent resolution of output images, rather than the ability to resolve - distinguish - features in the sample.

Response:

We would like to thank this reviewer who is correct that the physical or intrinsic resolution, governed by the imaging optics, is not improved by DNNs. What the DNNs improve are the spatial sampling density and SNR. The improved spatial sampling density enhances spatial feature visibility (or imaging resolution). We have revised the manuscript to have this point clearly stated. The terminology of "spatial resolution" has been changed to "imaging resolution" and highlighted in yellow in the revised manuscript.

Comment 2:

In this manuscript, the data being presented shows calcium transient events in labeled neurons. In all cases where calcium activity is strong and occupies the entire cell soma, estimated dF/F signals obtained by analysis of raw or enhanced data are about the same, and those events could have been detected without CNN enhancement, denoising does not unveil additional information.

My review aimed to identify what could be the benefits of the proposed image enhancement.

In the absence of simultaneously recorded high-resolution images, it is unclear if the dendrites identified in Figure 2d genuinely correspond to dendrites physically present at the depicted locations, or if they are only an interpretative display that best estimates where hypothetical dendrites should be placed to yield the acquired low resolution image. Similarly, the brief events that occur in speed enhanced outputs, are representations that may not be accurate. In other words, processing raw data through a DNN does not create new information, but reorganizes it in an output that appears to be of superior quality to the human eye. The proposed approach may have benefits, but only if the authors can make a case that enhancing the display with their algorithm enables the detection of features or events that are 1-verifiable with ground truth and 2 - not already visible in the raw data.

Response:

We thank the reviewer for these excellent questions.

(1) Key point 1 of the Comment: “Figure 2d genuinely correspond to dendrites physically present at the depicted locations, or if they are only an interpretative display that best estimates where hypothetical dendrites should be placed to yield the acquired low resolution image”

Response: We agree that the function of DNNs, generally speaking, is to make the best estimates. In our case, down-sampling enables a high imaging frame rate but it would also result in the loss of spatial features of a given object. The major goal of DNN-2 is to restore the spatial features (in addition to improving the image SNR).

(2) Key point 2 of the Comment: “The proposed approach may have benefits, but only if the authors can make a case that enhancing the display with their algorithm enables the detection of features or events that are 1-verifiable with ground truth and 2 - not already visible in the raw data.”

Response: Here is one example to show that image improved by DNN-2. The method enables detection of features which are not visible in the original (i.e., testing input) but verifiable with the ground truth:

The three columns shown in the figure (from left to right) are the original image (down-sampled *in vivo* image, here we chose a down-sampling factor $M=8$ as an example, corresponding to 64 spirals/frame), ground truth (*in vivo* 10-frame averaged image, collected at 512 spirals/frame), and the output image from the trained DNN-2, respectively. The testing images were collected from freely-moving mice. All the testing images were not used for DNN-2 training and DNN-2 did not have any priori information about those testing images. The details about generating the testing dataset and the evaluation of DNN-2 output are discussed in **Supplementary Information Figure S1**.

The figures in the first row show full-size images of one representative FOV. The figures in the second row show magnified views corresponding to the ROI marked in the white box. The soma feature was labeled as arrowhead 1, and the dendrite features were labeled as arrowheads 2 and 3, respectively. These morphology features were recognized based on their shape and size.

In the figures of the second row, the profile of a neuron soma (arrowhead 1) could be clearly resolved in the ground truth, but its edge became very blurred in the original image (i.e., the down-sampled image). Benefiting from the denoising and pixel inpainting functions of the trained DNN-2, the spatial profile of the soma was restored with much improved SNR in the DNN-2 output image.

Besides soma, the trained DNN-2 also helped restore other fine features such as dendrites (arrowheads 2 and 3) which remained consistent with the ground truth. The fine details could be clearly recognized in both the ground truth and the DNN-2 output images, but they were difficult to resolve in the testing input.

We have revised the manuscript accordingly. See Page 8, Lines 140-147:

“

To confirm the feasibility of the trained DNN-2, we manually selected and labeled a set of *in vivo* images collected from freely-behaving mice at a high scanning density (512 spirals/frame) as the testing dataset (see **Supplementary Information Figure S1** for more details) and quantitatively compared the quality of the DNN-2 output images with their corresponding *in vivo* ground truth. The results confirm that the as-trained DNN-2 works properly when applied to *in vivo* freely-behaving images with a low scanning density. The trained DNN-2 enabled detection of fine features that were consistent with the ground truth but difficult to resolve in the original images.

”

The restored imaging resolution and spatial features by the trained DNN-2 are very helpful for post-processing pipelines to correct image motion artifacts and segment neurons or dendrites.

Comment 3:

The manuscript in its current format should undergo major revisions, first to adequately classify novelty vs previously published work, then to better show the benefits of the proposed algorithm. The intrinsic resolution of the output images should be distinguished from the imaging resolution that characterizes the ability of the device to detect / separate physical features.

Response:

We appreciate the reviewer for summarizing his/her comments.

- 1) About novelty: See responses to General Comment 1.
- 2) About benefits: the proposed algorithm enables down-sampling and thus a higher imaging frame rate without compromising spatial features (or imaging resolution) or SNR. More examples are also presented in the Response to the specific Comment 2 (along with the associated benefits for image post-processing).
- 3) About resolution: As we discussed in the Response to Comment 1, the algorithm helps restore imaging resolution (not the physical or intrinsic resolution) and we have revised the manuscript accordingly.

Comment 4:

A scale bar would be most welcome in the supplementary videos to recall the dimensions of the field of view.

Response:

We are grateful to this reviewer for his/her kind suggestion. We have added a scale bar to the new version of the Supplementary videos.

References

1. C. Zhao, B. E. Dewey, D. L. Pham, P. A. Calabresi, D. S. Reich, and J. L. Prince, "SMORE: A Self-supervised Anti-aliasing and Super-resolution Algorithm for MRI Using Deep Learning," *IEEE transactions on medical imaging* (2020).
2. A. Krull, T. Vičar, M. Prakash, M. Lalit, and F. Jug, "Probabilistic noise2void: Unsupervised content-aware denoising," *Frontiers in Computer Science* **2**, 5 (2020).
3. K. Zhang, W. Zuo, Y. Chen, D. Meng, and L. Zhang, "Beyond a gaussian denoiser: Residual learning of deep cnn for image denoising," *IEEE transactions on image processing* **26**, 3142-3155 (2017).
4. H.-C. Park, H. Guan, A. Li, Y. Yue, M.-J. Li, H. Lu, and X. Li, "High-speed fiber-optic scanning nonlinear endomicroscopy for imaging neuron dynamics *in vivo*," *Opt Lett* **45**, 3605-3608 (2020).

Reviewers' Comments:

Reviewer #1:

Remarks to the Author:

The reviewer appreciates the great length with which the authors went to address the comments from last review. The reviewer believes, with the manuscript edits/additions and the newest supplementary information, that the authors have addressed the most critical comments put forth. The comments that have been "left to future work" are reasonable and understandable, thus do not warrant (in this reviewer's opinion) more exploration at this time.

All together, this reviewer recommends that the manuscript in its present form (excluding small grammatical errors) to be accepted for publication in Nature Communications.

Reviewer #2:

Remarks to the Author:

This reviewer is particularly disappointed by the response brought by the authors to major concerns.

My main concern is that the proposed method: using deep learning for denoising and upsampling in the spatial and temporal domain, certainly yields visually pleasing renderings and fluid videos showing calcium activity, but that, like any data processing techniques, it does not add more content to the raw data.

Therefore, it is always a possibility, that the enhanced output video data, denoised in the spatial domain and upsampled in the temporal domain, would either eliminate features and events that are contained in the raw data, or even create structures or events that do not physically exist. This is a common concern in any imaging technique where the data is processed before display.

Hence, to validate the proposed technique, the authors should show that the method they propose, improving images before segmenting them into calcium activity, is superior to existing techniques that process raw, noisy data sampled at speeds limited by the hardware capabilities.

Perhaps a good example (a recently published paper that postdates the manuscript and does not need to be cited <https://doi.org/10.1038/s41593-021-00895-5>) shows here, the output of a competing method, in parallel with electrophysiology data for validation. The ephys data serves as ground truth, and can be sampled at extremely high speeds.

In their response, the authors show example of structures that are hard to see in the raw data, but this is not sufficient to confirm that the proposed method does not erase or make up any neural activity.

At this point, While I am convinced that the proposed method has the benefit of yielding very eye-friendly video data of multiphoton calcium activity, and represents a convenient processing tool for qualitative renderings, The manuscript in its current format does not clearly show that the method proposed by the authors would outperform existing, predated calcium imaging analysis techniques (e.g. CaImAn, by Giovannucci et al) that were developed to process noisy data directly, without first denoising the images.

A potential reader of the manuscript may be inclined to believe that the sampling speed enhancement would be for instance compatible with the next generation of fast reporters of neural activity, e.g. voltage dyes, but the proposed method will still miss these fast voltage events, because slow hardware will not detect them, and no amount of data processing can make these events appear.

Despite significant improvements, the manuscript does not seem to introduce enough novelty to the field to reach a broad audience within the expectations of the community for Nature journals.

Detailed point-by-point responses:

Authors' responses are in blue, changes made to the manuscript in yellow, and all line numbers refer to the submitted documents.

Reviewers' Comments:

Reviewer 1

General comments:

The reviewer appreciates the great length with which the authors went to address the comments from last review. The reviewer believes, with the manuscript edits/additions and the newest supplementary information, that the authors have addressed the most critical comments put forth. The comments that have been "left to future work" are reasonable and understandable, thus do not warrant (in this reviewer's opinion) more exploration at this time.

Altogether, this reviewer recommends that the manuscript in its present form (excluding small grammatical errors) to be accepted for publication in Nature Communications.

Response:

We very much appreciate the reviewer's positive general comment on the innovation and potential impact of our work. We have carefully proofread the manuscript to improve/polish the language.

We would like to take this opportunity to thank this reviewer again for his/her positive comments in the last review on the novelty of the manuscript, particularly on our two-stage neural network training strategy when the ideal ground truth does not exist or is not readily available. We would also like to thank this reviewer for his/her detailed, constructive suggestions in the last review which helped guide our last revision and greatly improve the manuscript.

Reviewer 2

Comment 1:

This reviewer is particularly disappointed by the response brought by the authors to major concerns. My main concern is that the proposed method: using deep learning for denoising and upsampling in the spatial and temporal domain, certainly yields visually pleasing renderings and fluid videos showing calcium activity, but that, like any data processing techniques, it does not add more content to the raw data.

Therefore, it is always a possibility, that the enhanced output video data, denoised in the spatial domain and upsampled in the temporal domain, would either eliminate features and events that are contained in the raw data, or even create structures or events that do not physically exist. This is a common concern in any imaging technique where the data is processed before display.

Response:

We very much appreciate this reviewer's comments. For the last revision, we took this reviewer's comments very seriously and tried our best to carefully address them as detailed in the point-by-point responses in the last revision. Significant efforts were made to generate and analyze new data in order to address those critically important comments. We would like to take this valuable opportunity to refine some responses and address additional comments from this reviewer.

We would like to thank this reviewer for his/her comment on whether our method "eliminate features and events that are contained in the raw data, or even create structures or events that do not physically exist." We totally agree that this is a general concern literally for all image processing methods (traditional or deep-learning based ones).

(1) A commonly adopted criterion to address this concern is to compare the similarity between the our processed image and the ground truth [1]. In our last revision (**Supplementary Information Figure S1**), we provided the authenticity test of the trained DNN-2. We demonstrated that our trained DNN-2 enabled restoration of structural details which were consistent with the ground truth. We would like to emphasize that the ground truth we used in this test was the experimentally acquired ground truth rather than synthesized one. That test demonstrated that our proposed method will not change structural features.

(2) In addition, we also proved that the introduction of DNNs will not modify the temporal features in last revision (**Figure 2g**).

(3) To further address reviewer's concern, we performed the following test. The raw data was acquired from freely-behaving mice at an imaging frame rate of ~26fps which were used as the DNN-2 input. We then calculated the standard deviation projection map for the raw images and the corresponding DNN-2 processed ones. The value for each pixel in the projection map represents the standard deviation of the time series data at that given pixel. A higher value indicates more dynamic firing activities.

Standard Deviation Projection Map

As shown in the above figure, the standard deviation projection maps obtained from the raw images and the DNN-2 processed ones are highly consistent with a high structure similarity value MS-SSIM of ~ 0.93 . The high structure similarity between two standard deviation projection maps implies that our method maintains a high fidelity of the original raw images (and the neuron firing activities). This testing result has been added to the Supplementary Information (see **Figure S5**).

The above experimental evidence and analyses confirm that our proposed method is very unlikely to eliminate features/events or create features/events that do not physically exist in the raw data.

Comment 2:

Hence, to validate the proposed technique, the authors should show that the method they propose, improving images before segmenting them into calcium activity, is superior to existing techniques that process raw, noisy data sampled at speeds limited by the hardware capabilities.

Perhaps a good example (a recently published paper that postdates the manuscript and does not need to be cited <https://doi.org/10.1038/s41593-021-00895-5>) shows here, the output of a competing method, in parallel with electrophysiology data for validation. The ephys data serves as ground truth, and can be sampled at extremely high speeds.

In their response, the authors show example of structures that are hard to see in the raw data, but this is not sufficient to confirm that the proposed method does not erase or make up any neural activity.

At this point, While I am convinced that the proposed method has the benefit of yielding very eye-friendly video data of multiphoton calcium activity, and represents a convenient processing tool for qualitative renderings, The manuscript in its current format does not clearly show that the method proposed by the authors would outperform existing, predating calcium imaging analysis techniques (e.g. CaImAn, by Giovannucci et al) that were developed to process noisy data directly, without first denoising the images.

Response:

We appreciate the reviewer's comment. In addition to high image quality, we have performed new analyses and added new results to demonstrate other valuable advantages of our proposed image processing method, e.g., in (1) segmenting neurons with weak calcium signals which would otherwise be missed when using conventional method, and (2) identifying/segmenting finer structures such as dendrites for calcium signal analyses.

(1) The first example is about neuron segmentation. Our proposed method enables simultaneous denoising and pixel inpainting, and the improved image quality enables more reliable segmentation, especially for neurons with weak calcium signals. One representative comparison is shown below.

Here we chose the well-established neuron segmentation pipeline CaImAn (as suggested by the reviewer) for processing both the raw images and the ones processed by our method. The segmentation map shown in Figure (a-left) demonstrates that the pipeline missed one neuron (which had a weak calcium signal) when processing the raw image, while this neuron was recognized and segmented out by the pipeline when using our processed image (where the missed neuron was marked with dashed line as shown in Figure (a-right)). We further proved that the neuron truly exists rather than artifact, as shown in Figure (b) where the neuron exhibited spiking activities (see the time period marked in red as an example).

(2) Another example is about the segmentation of other fine structures such as dendrites, which can be better identified and segmented using well established methods from the processed images than the raw images. Noting that CaImAn does not work well for segmenting the fine structures thus here we adopted the max entropy thresholding method for segmentation [2]. One representative example is shown below:

Here the raw data was acquired at 26fps from a freely-behaving mouse (with one representative raw image shown in (a)). Then we applied well-established maximum entropy threshold method [3] to the raw image and the our DNN-2 processed image to generate segmentation maps. As shown in Figure (b), the segmentation map of the raw image exhibits noisy and severe discontinuities in dendrites. Conversely, the segmentation map of the DNN-2 processed image shows much clearer and more continuous dendritic profiles. Figure (c) shows the dynamic calcium signals for three representative dendritic ROIs, confirming the existence of dendrites (rather than noises or artifacts).

We have added the above discussion into the Supplementary Information (see **Figure S6** and **Figure S7**).

(3) Here we would also like to mention another benefit of our proposed method for motion artifact correction. It is well-known that a high SNR enjoys better motion correction performance with basically any motion correction pipelines such as Normcorr [4]. Although data are not shown here, we confirmed that our processing method improved motion artifact correction by using a simulated dataset. We would be delighted to include the results in the Supplementary Information if the reviewers and/or editors suggest so.

(4) Regarding comparison with electrophysiology, we did not conduct such a study. The connection between GCaMP signals and electrophysiology has been well studied which serves the foundation for GCaMP-based neural activity studies [5]. As we have shown that the processed images did not eliminate features/events nor create artificial ones and that this paper does not report new GCaMP-based neuronal functions, it is reasonable to believe that the comparison of the detected GCaMP signals with electrophysiology is beyond the scope of current manuscript.

Comment 3:

A potential reader of the manuscript may be inclined to believe that the sampling speed enhancement would be for instance compatible with the next generation of fast reporters of neural activity, e.g. voltage dyes, but the proposed method will still miss these fast voltage events, because slow hardware will not detect them, and no amount of data processing can make these events appear.

Response:

We fully understand the reviewer's concern about potential misunderstanding of our video rate (26 fps) imaging speed. This speed is obviously not sufficient to resolve action potential which would require a time resolution in millisecond level (corresponding to a frame rate of several hundred frames/second, which requires at least 10-fold frame rate increase over our current speed). We have added more discussions in the Discussion section to reiterate that the current frame helps mitigate motion artifacts and facilitates inferring some fast signals (such the firing rate or spikes [6]) yet it is not fast enough to resolve action potential signals.

Comment 4:

Despite significant improvements, the manuscript does not seem to introduce enough novelty to the field to reach a broad audience within the expectations of the community for Nature journals.

Response:

We respectfully disagree with this point of view. As reviewer 1 pointed out (and agreed by this reviewer as well), the method for generating the synthetic ground truth and the two-stage training strategy is novel and can be potentially useful for other scenarios where ideal ground truth is not readily available or does not exist. Our proposed method enables for the first time 10-fold frame rate improvement without sacrificing

the spatial imaging resolution and image signal to noise ratio. As discussed in Response to Comment 1, our proposed method is more sensitive to neurons of weak calcium signals and finer structures such as dendrites that are challenging to resolve with conventional methods. Along with these newly added analyses/results, we hope the reviewer would appreciate the innovation and novelty of this manuscript.

References

1. Z. Wang, A. C. Bovik, H. R. Sheikh, and E. P. Simoncelli, "Image quality assessment: from error visibility to structural similarity," *IEEE transactions on image processing* **13**, 600-612 (2004).
2. S. Basu, D. Plewczynski, S. Saha, M. Roszkowska, M. Magnowska, E. Baczynska, and J. Wlodarczyk, "2dSpAn: semiautomated 2-d segmentation, classification and analysis of hippocampal dendritic spine plasticity," *Bioinformatics* **32**, 2490-2498 (2016).
3. L. M. Martyushev and E. Axelrod, "From dendrites and S-shaped growth curves to the maximum entropy production principle," *Journal of Experimental and Theoretical Physics Letters* **78**, 476-479 (2003).
4. E. A. Pnevmatikakis and A. Giovannucci, "NoRMCorre: An online algorithm for piecewise rigid motion correction of calcium imaging data," *Journal of neuroscience methods* **291**, 83-94 (2017).
5. A. Song, J. L. Gauthier, J. W. Pillow, D. W. Tank, and A. S. Charles, "Neural anatomy and optical microscopy (NAOMi) simulation for evaluating calcium imaging methods," *Journal of Neuroscience Methods* **358**, 109173 (2021).
6. M. Pachitariu, C. Stringer, and K. D. Harris, "Robustness of spike deconvolution for neuronal calcium imaging," *Journal of Neuroscience* **38**, 7976-7985 (2018).

Reviewers' Comments:

Reviewer #1:

Remarks to the Author:

The reviewer would like to take a moment to address the comments (prior and most recent) of another reviewer. I will be referring to him/her as "**R2**". I will be referring to myself by "reviewer".

In their previous review, **R2** gave the following comment:

"The proposed approach to obtain the training data is innovative, however, the utilization of DNNs to denoise and upsample imaging data is well established in the computer vision community. Having seen several research papers utilize those methods for quantitative scientific imaging, *caution should be taken to adequately interpret the meaning of the output images.*"

... which was followed by this comment shortly after:

"In this manuscript, the data being presented shows calcium transient events in labeled neurons. In all cases where calcium activity is strong and occupies the entire cell soma, estimated dF/F signals obtained by analysis of raw or enhanced data are about the same, **and those events could have been detected without CNN enhancement, denoising does not unveil additional information.**"

And then, by this comment in the most recent review:

"Therefore, it is always a possibility, that the enhanced output video data, denoised in the spatial domain and upsampled in the temporal domain, would either eliminate features and events that are contained in the raw data, or even create structures or events that do not physically exist."

... And:

"In other words, the claims of resolution capabilities, often relate to the apparent resolution of output images, rather than the ability to resolve - distinguish - features in the sample."

This reviewer believes that the authors have addressed the concerns listed in these comments with a high degree of rigor. The authors' results from **Figure S1** are in line with the statement from **R2** (highlighted in yellow above) as to whether the CNN "**unveil[s] additional information**". To this reviewer, the best way to test an image reconstruction algorithm's biological applicability, especially an adversarial algorithm, is to ensure that the biological insight one can glean from it is in line with that of expected through controlled experiment. Hence, the reviewer believes that because the authors' algorithm does *not* introduce biological information that was not there previously, (illustrated in **Figure 2**, **Figure S1**, **Figure S2b**, **Figure S5**, **Figure S6** and **Figure S7**) and that the authors *never once claim to be able to*, the authors' argument is only strengthened.

Further, by gathering a second round of training data using the same imaging workflow but with a different cell type (astrocytes) and showing that their approach *will not* reconstruct data it is not trained for (**Figure S2**), the authors validate their algorithm is indeed not magic. To the reviewer, this addition was highly refreshing given the current state of publishing in deep learning applied to biological imaging (which is referred to by **R2** in their comment "**Having seen several research papers utilize those methods for quantitative scientific imaging, caution should be taken to adequately interpret the meaning of the output images**"). Caution *indeed should be taken by the authors herein* – a point which the authors have made certain to emphasize, and test quantitatively, throughout their manuscript.

Hence, this reviewer believes that **R2**'s argument that "... it is always a possibility, that the enhanced output video data, ..., would either eliminate features and events that are contained in the raw data, or

create structures or events that do not physically exist” is possibly made in bad faith. Again, the authors provided overwhelming evidence against this point. **R2** even states their unwillingness to budge on this point with their comment “In their response, the authors show example of structures that are hard to see in the raw data, but this is not sufficient to confirm that the proposed method does not erase or make up any neural activity”. This particular comment seems to suggest that *no* quantitative investigation would qualify as rigorous enough for **R2**’s standard. Despite this, the authors performed additional experiments/analyses to provide the requested evidence to support their presented method (**Figures S6 & S7**).

Further, by reading these two comments together, **R2** seems to simultaneously desire an algorithm that unveils new information (yellow highlight) and makes sure not to “... eliminate features ... or even create structures or events that do not physically exist”. I would like to emphasize that, *even though these comments by R2 contradict each other incredibly*, the authors have done an outstanding job at addressing the comments posed – in part through revisions that have strengthened their work remarkably (especially since the time of first submission).

--

With all the above said, perhaps it should be noted once more that the authors present and validate a highly novel data generation scheme that can be used on **countless data collection routines/imagers**. After reading this article > 1 year ago at the time of first submission, I was personally inspired greatly by the authors’ developed method. That inspiration was only strengthened (along with a sharp decrease in skepticism) by the revisions carefully provided by the authors after a great deal of effort. It just so happens that the authors also validated said DL workflow using an innovative technology and for a highly sought-after application. To this reviewer, this positions the authors’ work wonderfully for the audience of Nature Communications and only further supports my decision to accept as is.

Regardless of decision, this article will undoubtedly inspire a great deal of work once the rest of the scientific community gets the chance to read it.

Reviewer #2:

Remarks to the Author:

The authors have addressed my concerns in depth, and significantly improved the manuscript. Based on the enthusiasm of reviewer 1, I would like to mitigate my concerns of lack of novelty, and I am more inclined to support the manuscript for publication.

My only remaining concern is that the manuscript title, and abstract are slightly overstating the contents of the manuscript.

In my mind and in the mind of microscopists, the promise of a ten times speed enhancement means that I should expect the ability to resolve events that are ten times faster than my previous capability. For instance, with my current 2P microscope operating at 30 FPS, I am expecting, from the title and the abstract of this manuscript to be able to obtain an effective 300FPS imaging capability. Imaging capability. Can I reliably resolve rapid events that last, say 5 milliseconds? The answer is, as indicated in the author's response, that said event would still likely not be picked up in the raw data. If the event were to occur, say just after the laser beam scans their location, it will be over by the time the laser beam returns. If no trace of the event is present in the raw data, no amount of excellent software can make it appear. In other words, the proposed method cannot turn my system capturing images at 30fps setup into a 300 fps equivalent setup, only make reasonable guess of what to expect by leveraging spatial and temporal priors and expected redundancies in the specific case of neural data.

Therefore, with the intent not to set unreasonable expectations, the technique should not be presented as "imaging [...] with a ten-fold speed enhancement", but instead as a video-rendering technique, or a data processing technique, or any other appropriate wording that does not give the reader the impression that they should expect the same benefits one would get with a hypothetical 10X faster acquisition hardware.

Clearly, the method proposed by the authors has some advantages over existing state-of-the-art approaches such as Caiman, and the manuscript in its current version adequately documents these advantages. Since reviewer 1 seems to consider that these are sufficiently novel for publication in Nature Communications, I am happy to support this manuscript as well, if the authors are willing to phrase the description of their contributions in a way that sets more reasonable expectations.

Detailed point-by-point responses:

Authors' responses are in blue, changes made to the manuscript in yellow.

Reviewers' Comments:

Reviewer 1

General comments:

With all the above said, perhaps it should be noted once more that the authors present and validate a highly novel data generation scheme that can be used on countless data collection routines/imagers. After reading this article > 1 year ago at the time of first submission, I was personally inspired greatly by the authors' developed method. That inspiration was only strengthened (along with a sharp decrease in skepticism) by the revisions carefully provided by the authors after a great deal of effort. It just so happens that the authors also validated said DL workflow using an innovative technology and for a highly sought-after application. To this reviewer, this positions the authors' work wonderfully for the audience of Nature Communications and only further supports my decision to accept as is.

Regardless of decision, this article will undoubtedly inspire a great deal of work once the rest of the scientific community gets the chance to read it.

Response:

We very much appreciate the Reviewer 1's efforts in addressing some of the comments from Reviewer 2. We would like to take this opportunity to thank this reviewer again for his/her positive and constructive comments through the past two rounds of revisions.

Reviewer 2

General comments:

The authors have addressed my concerns in depth, and significantly improved the manuscript. Based on the enthusiasm of reviewer 1, I would like to mitigate my concerns of lack of novelty, and I am more inclined to support the manuscript for publication.

Response:

We would like to thank the reviewer for his/her support of the manuscript for publication.

Comment 1:

My only remaining concern is that the manuscript title, and abstract are slightly overstating the contents of the manuscript.

In my mind and in the mind of microscopists, the promise of a ten times speed enhancement means that I should expect the ability to resolve events that are ten times faster than my previous capability. For instance, with my current 2P microscope operating at 30 FPS, I am expecting, from the title and the abstract of this manuscript to be able to obtain an effective 300FPS imaging capability. Imaging capability. Can I reliably resolve rapid events that last, say 5 milliseconds? The answer is, as indicated in the author's response, that said event would still likely not be picked up in the raw data. If the event were to occur, say just after the laser beam scans their location, it will be over by the time the laser beam returns. If no trace of the event is present in the raw data, no amount of excellent software can make it appear. In other words, the proposed method cannot turn my system capturing images at 30fps setup into

a 300 fps equivalent setup, only make reasonable guess of what to expect by leveraging spatial and temporal priors and expected redundancies in the specific case of neural data.

Therefore, with the intent not to set unreasonable expectations, the technique should not be presented as "imaging [...] with a ten-fold speed enhancement", but instead as a video-rendering technique, or a data processing technique, or any other appropriate wording that does not give the reader the impression that they should expect the same benefits one would get with a hypothetical 10X faster acquisition hardware.

Response:

Following the reviewer's comment, we have revised the title and the abstract, making it clear that the 10x speed improvement is to video rate.

Comment 2:

Clearly, the method proposed by the authors has some advantages over existing state-of-the-art approaches such as Caiman, and the manuscript in its current version adequately documents these advantages. Since reviewer 1 seems to consider that these are sufficiently novel for publication in Nature Communications, I am happy to support this manuscript as well, if the authors are willing to phrase the description of their contributions in a way that sets more reasonable expectations.

Response:

We very much appreciate the reviewer's comments on the advantages and novelty of the methods reported in this manuscript. We have revised the title and the abstract to make it clear that the speed improvement was from about 2-3 frames/second to video rate (~26 frames/second).